# Secondary organic aerosol from chlorine-initiated oxidation of isoprene

Dongyu S. Wang and Lea Hildebrandt Ruiz

McKetta Department of Chemical Engineering, The University of Texas at Austin, Austin, Texas, USA

*Correspondence to*: Lea Hildebrandt Ruiz (lhr@che.utexas.edu)

**Abstract.** Recent studies have found concentrations of reactive chlorine species to be higher than expected, suggesting that atmospheric chlorine chemistry is more extensive than previously thought. Chlorine radicals can interact with $HO_x$ radicals and nitrogen oxides ($NO_x$) to alter the oxidative capacity of the atmosphere. They are known to rapidly oxidize a wide range of volatile organic compounds (VOC) found in the atmosphere, yet little is known about secondary organic aerosol (SOA)
formation from chlorine-initiated photo-oxidation and its atmospheric implications. Environmental chamber experiments were carried out under low-$NO_x$ conditions with isoprene and chlorine as primary VOC and oxidant sources. Upon complete isoprene consumption, observed SOA yields ranged from 7 to 36 %, decreasing with extended photo-oxidation and SOA aging. Formation of particulate organochloride was observed. A High-Resolution Time-of-Flight Chemical Ionization Mass Spectrometer was used to determine the molecular composition of gas-phase species using iodide-water and hydronium-water
cluster ionization. Multi-generational chemistry was observed, including ions consistent with hydroperoxides, chloroalkyl hydroperoxides, isoprene-derived epoxydiol (IEPOX) and hypochlorous acid (HOCl), evident of secondary OH production and resulting chemistry from Cl-initiated reactions. This is the first reported study of SOA formation from chlorine-initiated oxidation of isoprene. Results suggest that tropospheric chlorine chemistry could contribute significantly to organic aerosol loading.

## 1 Introduction

Studies have shown that long-term exposure to fine particulate matter (PM), also known as aerosol, is linked to increases in mortality and cardiorespiratory diseases (Dockery et al., 1993; Pope et al., 2006). Short-term exposure to aerosol could also induce stress response and cytotoxicity in cells (de Bruijne et al., 2009; Ebersviller et al., 2012; Hawley et al., 2014). Equilibrium partitioning of oxidized, semi-volatile organic compounds (Pankow, 1994), collectively referred to as secondary
organic aerosol (SOA), contributes 20–90 % to the global fine aerosol budget (Jimenez et al., 2009; Kanakidou et al., 2005). The majority of SOA originates from oxidation of biogenic volatile organic compounds (BVOCs), which account for ~90% of annual non-methane hydrocarbon emissions (Goldstein and Galbally, 2007; Guenther et al., 2012), among which isoprene has the highest emission rate at ~600 Tg $yr^{-1}$ (Guenther et al., 2006). Isoprene SOA formation initiated by ozone ($O_3$) , nitrate ($NO_3$), and hydroxyl (OH) radicals has been studied extensively and is estimated to account for 6–30 Tg $yr^{-1}$ of the global

aerosol budget (Brégonzio-Rozier et al., 2015; Claeys, 2004; Guenther et al., 2006; Kroll et al., 2006; Lin et al., 2012; Surratt et al., 2006, 2010; Zhao et al., 2015), but the importance of isoprene SOA formation within the marine boundary layer (MBL) remains highly disputed in literature (Arnold et al., 2009; Bikkina et al., 2014; Fu et al., 2011, 2013; Gantt et al., 2010, 2015; Hu et al., 2013; Luo and Yu, 2010; O'Dowd and de Leeuw, 2007). Although the production of reactive chlorine species such
as $Cl_2$/HOCl has been observed within the MBL (Lawler et al., 2011), little is known about SOA from chlorine-initiated oxidation of isoprene.

     Chlorine chemistry is known to have important effects on ozone layer depletion (Crutzen, 1974; Molina and Rowland, 1974). Recent laboratory studies and field measurements also suggest an important role of halogen (X) chemistry on tropospheric composition (Faxon and Allen, 2013; Finlayson-Pitts, 2010; Saiz-Lopez and von Glasow, 2012; Simpson et al.,
2015). Reactive halogen species in the form of $X_2$, XO, HOX, $XNO_2$, OXO are present in polar regions (Buys et al., 2013; Liao et al., 2014; Pöhler et al., 2010), the MBL (Lawler et al., 2011; Read et al., 2008), coastal and inland regions (Mielke et al., 2013; Riedel et al., 2012, 2013). Outside of the MBL and polar regions, natural emissions of reactive halogen species have been observed in volcano plumes (Bobrowski et al., 2007) and over salt lakes (Kamilli et al., 2016; Stutz, 2002). Anthropogenic sources include industrial emissions (Chang and Allen, 2006; Riedel et al., 2013; Tanaka et al., 2003), oil and gas production
(Edwards et al., 2014), water treatment (Chang et al., 2001), biomass burning (Lobert et al., 1999), engine exhaust (Osthoff et al., 2008; Parrish et al., 2009), and $NO_x$-mediated heterogenous reactions, notably the production of $ClNO_2$ via reactive uptake of $N_2O_5$ onto particles containing $Cl^-$ (Thornton et al., 2010). Recent studies have found that models under-predict the abundance of reactive halogen species, suggesting incomplete understanding of their sources (Faxon and Allen, 2013; Faxon et al., 2015; Simpson et al., 2015; Thornton et al., 2010). Photolysis of reactive halogen species produces halogen radicals that
can react with $O_3$, hydrocarbons, SOA, and other radicals in the atmosphere. Reactions with hydrocarbons and organic aerosols serve as chlorine and bromine radical sinks (Buxmann et al., 2015; Ofner et al., 2012; Platt and Hönninger, 2003), especially in high $NO_x$ environments where halogen recycling via $HO_x$ and XO reaction pathways is suppressed (Edwards et al., 2013; Riedel et al., 2014; Simpson et al., 2015).

     The concentration of chlorine radicals is estimated to be on the order of $10^2$ to $10^5$ molecules $cm^{-3}$ (Saiz-Lopez and von
Glasow, 2012; Wingenter et al., 1999, 2005), which is one to three orders of magnitude lower compared to OH radicals under ambient conditions (Faxon and Allen, 2013; Wingenter et al., 1999), but due to its high reactivity towards numerous VOCs, chlorine radicals could contribute significantly as a primary oxidant under certain conditions (Riedel et al., 2012; Riva et al., 2015; Young et al., 2014). Hydrogen-abstraction and chlorine-addition to VOC produce peroxy and chloroperoxy radicals, respectively, that could lead to the production of secondary $HO_x$ radicals and the formation of semi-volatile oxidized products.
Studies have shown that chlorine-initiated oxidation of alpha-pinene (Cai and Griffin, 2006; Ofner et al., 2013), toluene (Cai et al., 2008; Huang et al., 2014; Karlsson et al., 2001), and polycyclic aromatic hydrocarbons (PAHs, Riva et al., 2015) lead to SOA formation, with close to unity SOA yield reported for select PAHs (Riva et al., 2015). Reactive chlorine species could also enhance OH-radical propagation (Young et al., 2014), nocturnal $NO_x$ recycling (Riedel et al., 2012; Thornton et al., 2010), and ozone production (Tanaka et al., 2003), further increasing the oxidative capacity of the atmosphere.

Chlorine-initiated oxidation of isoprene could either proceed via the dominant (85 %) chlorine-addition pathway or a minor (15 %) hydrogen-abstraction pathway (Fantechi et al., 1998; Lei and Zhang, 2000; Nordmeyer et al., 1997; Orlando et al., 2003). Major gas-phase products include methyl vinyl ketone (MVK), methacrolein (MACR), chloroacetone, chloroacetaldehyde, hydrochloric acid, and isomers of 1-chloro-3-methyl-3-butene-2-one (CMBO), a unique tracer for
atmospheric chlorine chemistry (Nordmeyer et al., 1997; Riemer et al., 2008; Tanaka et al., 2003). The rate constant of the isoprene-chlorine reaction at 25°C ($2.64$–$5.50 \times 10^{-10}$ molecules$^{-1}$ cm$^3$) (Fantechi et al., 1998; Orlando et al., 2003; Ragains and Finlayson-Pitts, 1997) is much faster than the rate constant of the isoprene-OH reaction ($1.00 \times 10^{-10}$ molecules$^{-1}$ cm$^3$) (Atkinson and Arey, 2003), suggesting that isoprene-chlorine chemistry could compete with isoprene-OH chemistry under certain conditions. Moreover, reactions between chlorine and isoprene or isoprene-derived SOA could serve as a reactive
chlorine sink in the atmosphere, as has been proposed for reactions between chlorine and biogenic SOA (Ofner et al., 2012). To our knowledge, SOA formation from chlorine-initiated oxidation of isoprene has not been reported in literature. In order to address this significant knowledge gap, environmental chamber experiments were conducted using chlorine radicals as the primary oxidant source for isoprene oxidation. Experiments were conducted under low NO$_x$ conditions, using neutral or acidified seed aerosol to evaluate the effect of aerosol acidity on SOA formation. The mass yields and composition of SOA
formed from chlorine-initiated oxidation of isoprene are reported for the first time. Formation of organochlorides was also observed. Gas-phase measurements provide evidence of multi-generational oxidation chemistry, secondary HO$_x$ chemistry, and multifunctional gas-phase products.

## 2 Methods

### 2.1 Environmental Chamber Experiments

Experiments were performed at 25$^{\circ}$ C under low relative humidity (RH < 10 %) and low NO$_x$ (< 5 ppb) conditions in a 12 m$^3$ temperature-controlled Teflon$^{\circledR}$ chamber lined with UVA lights. Chamber characteristics were described elsewhere (Bean and Hildebrandt Ruiz, 2016). The UV spectrum is similar to other blacklight sources reported in literature (Carter et al., 2005). The NO$_2$ photolysis rate is used to characterize UV intensity and was determined to be 0.5 min$^{-1}$, similar to ambient levels (e.g. 0.53 min$^{-1}$ at 0 degrees zenith angle, Carter et al., 2005). Temperature, RH, and the concentrations of O$_3$, NO, NO$_2$, and
NO$_x$ were continuously monitored. The chamber was flushed with dry clean air generated by a clean air generator (Aadco, 737R) at a flowrate exceeding 100 liters per minute (LPM) for at least 12 hours before each experiment. Between experiments, "blank experiments" were conducted in which seed particles, O$_3$, and Cl$_2$ (Airgas, 106 ppm in N$_2$) were injected into the chamber at high concentrations and UV lights were turned on to remove any residual organics which could be released from the Teflon$^{\circledR}$ chamber surface. Background effects have been quantified using chamber characterization experiments (Carter et
al., 2005) and the SAPRC chamber modeling software (http://www.engr.ucr.edu/~carter/SAPRC/) in combination with the Carbon Bond 6 (CB6r2) chemical mechanism, which was modified to include basic gas phase inorganic chlorine chemistry in addition to Cl$_2$ and ClNO$_2$ photolysis (Sarwar et al., 2012; Yarwood et al., 2010). NO$_x$-offgasing is represented within the

model by a constant emission of nitrous acid (HONO) from the chamber walls on the order of 0.1 ppb min[-1], which was determined separately in chamber characterization experiments (Carter et al., 2005).

For each experiment, microliters of isoprene (Acros Organics, 98 % stabilized) were transferred into a glass sampling tube (Kimble-Chase, 250 ml), which was then flushed with clean air into the chamber. Depositional particle wall loss was

constrained using dried, monomodal, and polydisperse seed particles introduced prior to photo-oxidation using an Aerosol Generation System (Brechtel, AGS Model 9200). Neutral seed particles were generated from a 0.01 M ammonium sulfate solution; acidic seed particles were generated using a solution containing 0.005 M ammonium sulfate and 0.0025 M sulfuric acid. Two $Cl_2$ injection methods were used: For "initial $Cl_2$" experiments, all $Cl_2$ was injected in the dark and allowed to mix with isoprene prior to photo-oxidation. For "continuous $Cl_2$" experiments, $Cl_2$ was injected continuously with UV lights on at

0.1 LPM with an additional 0.9 LPM clean air dilution flow, equivalent to ~0.88 ppb min[-1] $Cl_2$ into the chamber. Initial $Cl_2$ experiments were performed to achieve rapid oxidation of isoprene and to separate initial SOA formation from effects of vapor wall loss. Because chlorine radicals are not expected to regenerate, continuous $Cl_2$ injection experiments were performed to provide more steady but lower chlorine radical concentrations.

## 2.2 Instrumentation

A High-Resolution Time-of-Flight Chemical Ionization Mass Spectrometer (Aerodyne Research Inc., "CIMS") was used to measure gas-phase organic compounds. Detailed theory and operation of the CIMS are discussed elsewhere (Aljawhary et al., 2013; Bertram et al., 2011; Lee et al., 2014; Yatavelli et al., 2012). Reagent ions are generated by passing humidified UHP $N_2$ over a methyl iodide permeation tube and then through a [210]Po radioactive cartridge (NRD, P-2021) at 2 LPM into the ion-molecule reaction (IMR) chamber operating at 200 mbar pressure. Analyte, "M" can undergo chemical ionization within the

IMR with hydronium-water, $(H_2O)_n(H_3O)^+$ or iodide-water, $(H_2O)_nI^-$ ion clusters,

$$(H_2O)_n(H_3O)^+ + M \rightarrow (H_2O)_n(MH)^+ + H_2O \tag{1}$$

$$(H_2O)_nI^- + M \rightarrow (H_2O)_n(MI)^- \tag{2}$$

where the number of clusters, "n" ranges from 0 and 2 for $(H_2O)_n(H_3O)^+$, with $(H_2O)(H_3O)^+$ being the most dominant reagent ion in the positive ion mode. Hydronium-water cluster CIMS was used to detect isoprene and select moderately oxidized

species (Aljawhary et al., 2013). For $(H_2O)_nI^-$ ionization, "n" ranges from 0 to 1, with $I^-$ being the most dominant reagent ion. Water-iodide cluster CIMS was used to detect select highly oxidized and acidic species (Aljawhary et al., 2013; Lee et al., 2014). An Aerosol Chemical Speciation Monitor (Aerodyne Research Inc., "ACSM") was used to determine the bulk chemical composition of submicron, non-refractory aerosol species (Ng et al., 2011a). Analytes are flash-vaporized at 600° C, ionized via electron impact ionization (EI), and then measured by a quadrupole mass spectrometer (Ng et al., 2011a). Background-

corrected ("difference") mass spectra are obtained by subtracting filtered ("closed") from unfiltered ("open") measurements (Ng et al., 2011a). A standard fragmentation table is used to speciate difference mass spectra (Allan et al., 2004). Calibration is performed with 300 nm size-selected ammonium nitrate and ammonium sulfate aerosol to determine the response factor for

particulate nitrate and the relative ionization efficiencies (RIE) for sulfate and ammonium; these values are needed to convert ion intensities to mass concentrations (Ng et al., 2011a). Particle volume and size distributions were measured using a Scanning Electrical Mobility Spectrometer (Brechtel, SEMS Model 2002) consisting of a Differential Mobility Analyzer and a butanol Condensation Particle Counter. SEMS sheath and sample flowrates were set to 5 and 0.35 LPM, respectively, covering a 10–1000 nm sizing range.

## 2.3 Data Analysis

Suspended particles are lost to the Teflon® chamber wall over time, for which numerous correction methods have been proposed (Carter et al., 2005; Hildebrandt et al., 2009; Nah et al., 2017; Ng et al., 2007; Pathak et al., 2007; Pierce et al., 2008; Verheggen and Mozurkewich, 2006). Recent studies also report loss of organic vapors to Teflon® surfaces (Kokkola et al., 2014; Krechmer et al., 2016; Loza et al., 2010; Matsunaga and Ziemann, 2010; Zhang et al., 2015). Assuming internal mixing of particles and that organic vapor can condense onto suspended and wall-deposited particles alike, we corrected for particle wall loss and the loss of organic vapors onto wall-deposited particles using the organic-to-sulfate ratio (Hildebrandt et al., 2009),

$$C_{OA}(t) = \frac{c_{OA}^{sus}(t)}{c_{seed}^{sus}(t)} C_{seed}^{sus}(t = 0) \tag{3}$$

where $C_{OA}^{sus}(t)$ is the suspended organic aerosol (OA) concentration, which was zero at the start of each experiment, $C_{seed}^{sus}(t)$ is the suspended seed particle concentration, $C_{seed}^{sus}(t = 0)$ is the suspended seed particle mass concentration at the start of photo-oxidation, and $C_{OA}(t)$ is the corrected OA concentration. This correction does not account for loss of organic vapors to clean Teflon® surfaces.

SOA yield, $Y$, is calculated as,

$$Y = \frac{C_{OA}}{\Delta VOC} \tag{4}$$

where $\Delta VOC$ is the amount of VOC consumed. Based on absorptive equilibrium partitioning theory (Odum et al., 1996; Pankow, 1994), the volatility basis set (VBS) framework (Donahue et al., 2006) states that

$$Y = \xi = \sum_i \alpha_i \xi_i \tag{5}$$

$$C_i = \alpha_i \Delta VOC \tag{6}$$

$$\xi_i = \left(1 + \frac{C_i^*}{C_{OA}}\right)^{-1} \tag{7}$$

$$C_{OA} = \sum_i C_i \xi_i \tag{8}$$

where $C_i^*$ is the effective saturation concentration of the surrogate compound in VBS bin $i$ in μg m⁻³; $\alpha_i$, $C_i$, $\xi_i$ are the total yield, total mass concentration, and the condensed phase mass fraction of bin $i$, respectively. By rearranging Eq. (5)-(8), an expression for the minimum VOC consumption required for SOA formation can be derived (see section S1),

$$(\Delta VOC_{\min})^{-1} = \sum_1^n \frac{\alpha_i}{C_i^*} \tag{9}$$

where $\Delta VOC_{\min}$ is low if low-volatility compounds dominate the aerosol phase. When aerosol volatility is low and aerosol loading is high, such that the condensed phase mass fraction, $\xi_i$ from Eq. (7) approaches 1, the "maximum" SOA yield (Griffin et al., 1999) is reached, where

$$Y_{\max} = \sum_1^n \alpha_i \tag{10}$$

The extent of SOA oxidation is depicted using $f_{44}$ and $f_{43}$, which represent the fractional contribution to the total organic ion signal measured by the ACSM from ion fragments at mass-to-charge ($m/z$) 44 (mostly $CO_2^+$, a proxy for doubly-oxidized compounds) and at $m/z$ 43 (mostly $C_2H_3O^+$, a proxy for singly-oxidized compounds), respectively (Chan et al., 2010; Chhabra et al., 2011; Ng et al., 2011b). Based on empirical correlations, the oxygen-to-carbon ratio (O:C), the hydrogen-to-carbon ratio (H:C), and the oxidation state of carbon ($\overline{OS}_c$) can be estimated from $f_{44}$ alone as summarized in section S2 (Canagaratna et al., 2015; Donahue et al., 2012; Heald et al., 2010; Kroll et al., 2011). The empirical correlations were derived using a comprehensive collection of Aerosol Mass Spectrometer datasets but may underestimate O:C values for SOA formed under low NO$_x$ conditions from isoprene or toluene (Canagaratna et al., 2015). Variability in $f_{43}$ and $f_{44}$ among different ACSMs has also been reported (Crenn et al., 2015).

High resolution mass spectra fitting of CIMS data was performed using the software Tofware V2.5.7 (Tofwerk) in Igor Pro V6.37 (Wavemetrics Inc.,). CIMS sensitivity correction utilized the Active Chemical Ionization Mass Spectroscopy (ACIMS) formula (de Gouw and Warneke, 2007), normalizing all product ion signals against dominant reagent ion signals, $(H_2O)_nH_3O^+$ and $(H_2O)_nI^-$. For most experiments conducted, the reagent signals were at least five times greater than the summed product signals, equivalent to less than 10 % overcorrection by ACIMS compared to more rigorous methods such as Parallel-ACIMS, which accounts for reagent ion depletion (see section S3 for a more detailed discussion). Because the CIMS cannot distinguish between isomers and because of the lack of calibration standards, gas-phase data presented here are normalized by the maximum signals for better visualization and evaluation of qualitative trends.

## 3 Results and Discussion

### 3.1 SOA Formation, Aging, and Composition

Table 1 summarizes the experimental conditions and results. Figure 1 compares wall loss-corrected SOA time series from three experiments with similar precursor concentrations. In the continuous Cl$_2$ experiments (C2 and C7), isoprene gradually reacted away during Cl$_2$ injection; SOA concentration steadily increased until isoprene was depleted, at which point SOA

concentration began to decay. The decay of SOA was likely due to oxidative fragmentation (Kroll et al., 2011), vapor wall loss (Boyd et al., 2015; Krechmer et al., 2016), and/or photolysis of, for example, peroxide species (Kroll et al., 2006; Surratt et al., 2006), which were observed in the gas-phase (see Section 3.4). The initial $Cl_2$ experiment (A4) exhibited similar trends, but the SOA decay was faster, where 30 $\mu$g m$^{-3}$ SOA decay (40 % of maximum SOA mass) occurred within about 30 minutes

of photo-oxidation (9 < T < 40 min), likely due to more rapid oxidation and fragmentation of reaction products. Under UV, the effects of oxidative fragmentation and photolysis cannot be separated from the effects of vapor wall loss. SOA decay was slower in the dark than under UV (see section S2 and Fig. S1). Vapor wall effects are expected to be more important when the concentrations of the oxidant and OA are lower, in which case oxidative fragmentation effects are weaker and less absorbing mass is available to compete with wall loss. After extended photo-oxidation (T > 100 min), SOA concentrations achieved via

initial $Cl_2$ injection (Exp. A4) and continuous $Cl_2$ injection (Exp. C2) differed by less than 8 $\mu$g m$^{-3}$ (< 20 % of total SOA mass at T = 100 min for Exp. C2). The data shown in Fig. 1 and summarized in Table 1 also suggest that aerosol acidity promotes SOA formation: The SOA concentrations observed in acidified seed Exp. C7 were more than twice as high as SOA concentrations observed in neutral seed Exp. C2. In addition, greater seed aerosol surface area appears to result in higher SOA concentrations, as shown in Table 1 for Exp. A6 and Exp. A7. The formation of SOA began shortly after UV lights were turned

on in all experiments. $\Delta VOC_{min}$ from Eq. (9) is therefore small, suggesting that the initial oxidation products responsible for SOA formation have very low volatility. Prompt SOA formation was also observed in previous work for chlorine-initiated oxidation of alpha-pinene (Cai and Griffin, 2006; Ofner et al., 2013) and toluene (Cai et al., 2008; Huang et al., 2014; Karlsson et al., 2001). Formation of low volatility early generation products may be a common feature of chlorine-initiated oxidation.

Figure 2 shows that SOA oxidation state, represented by $f_{44}$ and estimated $\overline{OS_c}$, depended on the initial isoprene

concentration and was unaffected by the oxidant injection method. Because isoprene is more reactive towards chlorine radicals than its oxidation products, such as MVK and MACR (Orlando et al., 2003), isoprene could scavenge radicals and delay SOA aging. Additionally, increased OA mass loading could absorb less oxidized and more volatile compounds into the particle-phase, lowering the observed SOA oxidation state at higher SOA loadings resulting from higher initial isoprene concentrations. The estimated $\overline{OS_c}$ of chlorine-isoprene SOA increased from -0.5 to over 1 during Exp. C1, characteristic of the evolution of

semi-volatile oxygenated OA (SV-OOA) to low-volatility oxygenated OA (LV-OOA) (Kroll et al., 2011). Oxidation of chlorine-isoprene SOA formed under low NO$_x$ follows a similar trajectory as OH-isoprene SOA formed in previous work under higher NO$_x$, which is considerably more oxidized than OH-isoprene SOA formed under low NO$_x$ (Chhabra et al., 2011), as shown in Fig. 3. The oxidizing capacity of chlorine radicals has also been demonstrated for select biogenic SOA derived from alpha-pinene, catechol, and guaiacol, where halogenation led to significant SOA aging, formation of high molecular

weight compounds, and particle growth (Ofner et al., 2012). High reactivity of chlorine radicals towards isoprene and its reaction products meant that extensive SOA processing could be easily achieved within laboratory timescales.

## 3.2 Particulate Organochloride

Chlorine addition to the double bond is the dominant (~85%) isoprene-chlorine reaction pathway (Fan and Zhang, 2004; Ragains and Finlayson-Pitts, 1997), and thus the formation of semi-volatile and low-volatility chlorinated organic compounds would be expected. Aerosol analytes undergo electron impact ionization in the ACSM, and chlorine-containing ion fragments are mostly expected at *m/z* 35 and 37 as $Cl^+$ and at *m/z* 36 and 38 as $HCl^+$. Larger organochloride ion fragments may exist but cannot be separated in the unit-mass resolution spectra. A previous study on chorine-initiated oxidation of toluene, which proceeds primarily through a hydrogen-abstraction pathway, reported particulate chlorine formation (4 % of the total aerosol mass), which was attributed to HCl uptake (Cai et al., 2008). Formation of organochloride aerosol has been observed previously for chlorine-initiated oxidation of alpha-pinene (Ofner et al., 2013), which proceeds primarily via a chlorine-addition reaction pathway. Thus, isoprene-chlorine reactions are expected to result in particulate organochloride formation. The uptake of HCl produced from hydrogen-abstraction or intramolecular HCl elimination (Ragains and Finlayson-Pitts, 1997) could also contribute slightly to observed particulate chlorine. Organochloride has also been observed in biogenic SOA post-processed by halogenation (Ofner et al., 2012), in new particles formed from 1,8-cineol and limonene over simulated salt lakes (Kamilli et al., 2015), and in-situ over salt lakes (Kamilli et al., 2016). To date, this is the only reported study of organochloride measurement using an ACSM.

As shown in Fig. 4 for Exp. A8, the ACSM initially observed significant levels of particulate chlorine (over 9 % of the total SOA mass), which then decreased to near background levels. In other experiments, particulate chlorine concentrations were near the detection limit. The low observed chlorine concentration was likely due to incomplete vaporization of chlorinated compounds during the sample ("open") period, resulting in particulate matter vaporization during the filter ("closed") period and overestimation of background signals, as explained in more detail in section S4. Further analysis shows that while the difference signal of a faster-desorbing chlorine ion fragment ($HCl^+$) correlates well with OA, the difference signal of a slower-desorbing ion fragment ($Cl^+$) anti-correlates with OA (Fig. S7). The background $Cl^+$ signal was consistently higher than the sample $Cl^+$ signal, except when the sample organochloride concentration increased much faster than background, as it does during the SOA formation period in initial $Cl_2$ injection experiments. Outside the initial SOA formation period, difference $Cl^+$ signals were negative and difference $HCl^+$ signals were positive, the summation of which resulted in the low apparent particulate chlorine concentrations.

Low abundance has been cited as the reason for highly variable measurements of ambient particulate chlorine using ACSM and similar instruments (Crenn et al., 2015). Overall, our results indicate that the standard operating procedure of the ACSM and similar instrumentation may systematically underestimate particulate chlorine concentration. To better quantify particulate chlorine, the filtered measurement period could be extended to better capture the true background. Higher vaporizer temperatures could be applied to desorb low-volatility species more efficiently, but doing so could also change the fragmentation profile of all aerosol components. Another approach would be to only use fast-desorbing ions ($HCl^+$) for

quantification: For SOA from chlorine-initiated oxidation of isoprene under low $NO_x$, the average $(HCl^+)$-to-organics ratio was $0.07 \pm 0.01$ (Fig. S8).

## 3.3 SOA Yield

It is customary to present SOA yield as a function of OA loading. From Eq. (4), when the VOC precursor has been depleted, all subsequent yield $Y$ varies linearly with $C_{OA}$ along a slope of $(\Delta VOC_0)^{-1}$, the inverse of the initial VOC concentration. Post VOC depletion, SOA mass may further increase as multi-generational oxidation products partition to the particle phase, but SOA mass will eventually decrease due to fragmentation (Kroll et al., 2011). In theory, all VOC-oxidant mixtures whose $\Delta VOC_{min}$ is less than $VOC_0$ eventually fall somewhere on the same, pre-defined yield "line" with a slope of $(\Delta VOC_0)^{-1}$, and will converge (to Cartesian origin) over time as fragmentation continues. This pre-defined yield curve is thus non-unique and depends only on $VOC_0$. Incorporating data collected after VOC depletion, whether from the same experiment or different experiments with similar initial $VOC_0$ values, biases the VBS fitting parameters towards the pre-defined yield curve (see section S5). Therefore, one-dimensional VBS fitting is not sufficient to describe SOA formation and oxidation post VOC depletion, as previous studies have noted (Kroll et al., 2007; Liu et al., 2016; Xu et al., 2014). Two-dimensional modeling would be more appropriate in these cases (Chuang and Donahue, 2016; Donahue et al., 2012; Murphy et al., 2012) but was not performed on this dataset.

Complete isoprene depletion, which coincides with the maximum SOA concentration (see Fig. 1), is used as the reference condition for yield reporting in Table 1. Due to uncertainties with organochloride quantification, the particulate chlorine content was not included in the SOA yield calculation. Only results from neutral seed experiments are used for comparison with other literature values of OH-oxidation under low and high $NO_x$ scenarios (Brégonzio-Rozier et al., 2015; Koo et al., 2014; Liu et al., 2016; Xu et al., 2014) in Fig. 5. In initial $Cl_2$ injection experiments, maximum SOA concentrations were reached within 15 minutes, and the effects of vapor wall loss, oxidative fragmentation, and photolysis on reported maximum SOA yields were lower than during continuous $Cl_2$ injection experiments. Observed SOA yields averaged $20 \pm 3$ % for initial $Cl_2$ injection cases (A1–A5) and $8 \pm 1$ % for continuous $Cl_2$ injection cases (C1–C4). Under atmospheric conditions, the isoprene-to-chlorine ratio will usually be higher than ratios used in these experiments (0.5-1.2). Previous studies on chlorine-initiated SOA formation from toluene (Cai et al., 2008) and limonene (Cai and Griffin, 2006) suggest that SOA yields decrease with higher VOC-to-chlorine ratio. While we did not observe a clear correlation between SOA yield and isoprene-to-chlorine ratios used here (0.5-1.2), such dependence could be present over a wider ratio range. For air quality models which do not explicitly account for fragmentation reactions, the use of the average continuous case yield ($8 \pm 1$ %), which is similar to recently reported OH-oxidation yields (Liu et al., 2016; Xu et al., 2014), is more appropriate because the isoprene-to-chlorine ratio is closer to atmospheric conditions and because the SOA yields from continuous injection experiments account for effects of OA aging in the atmosphere (which occur throughout the experiments). The presence of acidic aerosols and inclusion of particulate chlorine content would increase expected yields.

### 3.4 Gas-phase products: Multigenerational oxidation, organochloride formation and secondary OH chemistry

Using $(H_2O)_nH_3O^+$ CIMS, we observed ions consistent with isoprene ($C_5H_8^+$ and $(C_5H_8)H^+$), MACR/MVK ($(C_4H_6O)H^+$), isomers of CMBO ($(C_5H_7ClO)H^+$), 2-methyl-3-buten-2-ol (MBO) ($(C_5H_6O)H^+$), chloroacetone ($(C_3H_5ClO)H^+$), and other gas-phase oxidation products, as shown in Fig. 6a. CMBO, MBO, and MVK/MACR were among the earliest oxidation products,
which were further oxidized to produce SOA. MVK and MACR are also key intermediary products in OH-isoprene reactions from NO + $RO_2$ and $RO_2$ + $RO_2$ reaction pathways, where MACR is a known SOA precursor (Brégonzio-Rozier et al., 2015; Kroll et al., 2006; Surratt et al., 2006; Xu et al., 2014), which could contribute to some similarities between OH-isoprene and chlorine-isoprene SOA. Multiple generations of chlorinated $C_5$ compounds were observed in the gas-phase, as shown in Fig. 6a and 6b, possibly from continued oxidation of CMBO and MBO. Another gas-phase product, $C_5H_8O$ also appeared to act as
a SOA precursor, and ions resembling its oxidation products (e.g., $C_5H_7ClO_2$ and $C_5H_9ClO_3$) were observed (see Fig. S12). The formation of $C_5H_8O$ from chlorine-isoprene oxidation has been proposed in literature (Nordmeyer et al., 1997). Previous studies that observed $C_5H_8O$ during OH-initiated oxidation of isoprene identified it as 2-methylbut-3-enal, but its formation mechanism remains unclear (Brégonzio-Rozier et al., 2015; Healy et al., 2008) and is beyond the scope of this work. It should be noted that CMBO has long been identified as a unique gas-phase marker for isoprene-chlorine oxidation (Chang and Allen,
2006; Nordmeyer et al., 1997; Riemer et al., 2008; Tanaka et al., 2003). Degradation of CMBO by OH is implemented in some air quality models, though only with inferred reaction rates (Tanaka and Allen, 2001). Our gas-phase measurements and proposed reaction products suggest that CMBO undergoes further oxidation reactions initiated by chlorine. This degradation of CMBO, and its role as a potential SOA precursor, could have important implications for estimating atmospheric chlorine activity and warrants further investigation.

The formation of CMBO and MBO produces $HO_2$ radicals (Orlando et al., 2003; Ragains and Finlayson-Pitts, 1997), which serve as a source of secondary OH radicals. Other $RO_2$ + $RO_2$ reaction pathways also produce $HO_x$ radicals. Evidence of secondary OH radical production has been reported for $NO_3$-oxidation of isoprene (Kwan et al., 2012; Schwantes et al., 2015), for chlorine-initiated oxidation of methylnaphthalenes and naphthalene under low-$NO_x$ (Riva et al., 2015; Wang et al., 2005) and for chlorine-initiated oxidation of toluene under high $NO_x$ (Huang et al., 2012). Figure 6b shows the accumulation of
HOCl observed using $(H_2O)_nI^-$ CIMS during the SOA growth period, where $HO_x$ radicals produced from chlorine-isoprene oxidation could react with excess chlorine. Production of OH is also possible from reactions between NO and $HO_2$ radicals, which would be more pronounced under high $NO_x$ conditions. No $NO_x$ was added to our experiments, and measured concentrations were below 5 ppb, mostly in the form of $NO_2$, which may have been released from the Teflon® surface under UV. We use the SOA trend as a common reference because isoprene is not detected using $(H_2O)_nI^-$ CIMS. It can be seen in
Fig. 1 and 6a that isoprene depletion roughly coincides with the SOA concentration peak. This explains the reversal from HOCl production to HOCl decay following the SOA concentration peak shown in Fig. 6b, when $HO_x$ radical production from isoprene-chlorine oxidation (e.g. formation of CMBO and MBO) ceased due to isoprene depletion. Accumulated HOCl was then gradually photolyzed under UV. In the absence of a primary oxidant source, such as when chlorine injection stopped in

Exp. C3, HOCl could provide residual OH and chlorine radicals under UV. Under dark, dry, and low-$NO_x$ conditions, HOCl remains stable as a temporary radical reservoir, as shown in Fig. 6b during the period when UV lights were turned off.

Secondary OH chemistry also may have contributed to SOA formation. For instance, the $C_5H_8ClO_2{}^\bullet$ radical produced via chlorine-addition to isoprene could either undergo $RO_2 + RO_2$ chemistry to produce $C_5H_7ClO$ (e.g. CMBO) or undergo $RO_2 +$ $HO_2$ chemistry to produce $C_5H_9ClO_2$, a chloroalkyl hydroperoxide. Similarly, the $C_5H_7O_2{}^\bullet$ radical produced via hydrogen-abstraction from isoprene could either undergo $RO_2 + RO_2$ chemistry to produce $C_5H_6O$ (e.g. MBO) or undergo $RO_2 + HO_2$ chemistry to produce $C_5H_8O_2$, a hydroperoxide. Ions consistent with $C_5H_9ClO_2$ and $C_5H_8O_2$ were observed in the gas-phase, as shown in Fig 6a and 6b, where the formation of $RO_2 + HO_2$ reaction products appeared delayed compared to their $RO_2 +$ $RO_2$ pathway counterparts. This is consistent with $RO_2 + RO_2$ reactions being a source of $HO_x$ radicals. Figure 7 summarizes select observed gas-phase products whose time series are shown in Figure 6a and 6b. More detailed reaction pathways and time-series comparisons are shown in Figures S10-12 in section S6. In the OH-isoprene system, multifunctional, low volatility hydroperoxides produced from non-IEPOX ("isoprene-derived epoxydiol") reaction pathways contribute to SOA formation under low $NO_x$ conditions (Krechmer et al., 2015; Liu et al., 2016; Riva et al., 2016). Analogously, in the Cl-isoprene system, the (chloroalkyl) hydroperoxide species identified in Fig. 6 and Fig. 7 are expected to contribute to SOA formation.

The ion consistent with isoprene-derived hydroxyl hydroperoxides (ISOPOOH) or IEPOX, $C_5H_{10}O_3$, was observed in $(H_2O)_nI^-$ CIMS, as shown in Fig. 6b. Reactive uptake and oxidation of IEPOX has been reported to contribute significantly to SOA mass during isoprene OH-oxidation, especially under acidic or humid conditions (Bates et al., 2014; Gaston et al., 2014; Lewandowski et al., 2015; Lin et al., 2012; Nguyen et al., 2014; Paulot et al., 2009; Surratt et al., 2010). For chlorine-initiated oxidation of isoprene, the SAPRC chamber model results indicate that over 99% of the isoprene reacts with chlorine; OH oxidation of isoprene is therefore only a very minor pathway in these experiments. Model results also show that $HO_2$ production is dominated by isoprene-chlorine chemistry when sufficient isoprene is present, whereas wall effects dominate $HO_2$ production ($> 60\,\%$) after all isoprene has been consumed. The model does not explicitly represent chlorine-initiated oxidation of reaction products, which can produce additional $HO_x$ radicals, and therefore likely underestimates the importance of secondary OH chemistry. Furthermore, we note that the observed $(C_5H_{10}O_3)H^+$ and $(C_5H_{10}O_3)I^-$ do not necessarily correspond to ISOPOOH or IEPOX. For instance, whereas $(C_5H_{10}O_3)H^+$ in Fig. 6a trended with $(C_5H_8O_2)H^+$ during SOA growth, $(C_5H_{10}O_3)I^-$ in Fig. 6b appeared early on as an SOA precursor and exhibited a different qualitative trend than $(C_5H_8O_2)I^-$. We can therefore infer that $(C_5H_{10}O_3)H^+$ is an ion cluster in the form of $(C_5H_8O_2)(H_3O)^+$. Without any information on the chemical structure, it is also possible that $(C_5H_{10}O_3)I^-$ is the fragment of some unidentified parent ion(s) or that it is an IEPOX isomer produced from non-OH-reaction pathways. Moreover, photooxidative degradation of chlorinated organic compounds could produce products that resemble OH-oxidation products like $C_5H_{10}O_3$. A similar observation has been reported for chlorine-initiated oxidation of alpha-pinene, where the SOA appeared less like halogenated organic aerosol as oxidation continued (Ofner et al., 2013).

Another way to test the presence of IEPOX is to reduce aerosol pH, which should lead to increased uptake of IEPOX (Budisulistiorini et al., 2013; Gaston et al., 2014; Hu et al., 2015; Lin et al., 2012; Riedel et al., 2015, 2016). Comparison of

ACSM mass spectra (see Fig. S13) suggests that the presence of acidic aerosol increases the contribution of ion mass fragments at $m/z$ 82 ($C_5H_6O^+$, "$f_{82}$") to the overall SOA mass, which is associated with IEPOX-derived OA (Budisulistiorini et al., 2013; Hu et al., 2015). However, the magnitude of change is low (1 ‰) and within uncertainty of the instrument. Interference by non-IEPOX-derived OA fragments and non-$C_5H_6O^+$ ions at $m/z$ 82 is also possible. Separate monoterpene-chlorine experiments observed $f_{82}$ values as high as 5 ‰. The observed $f_{82}$ values for isoprene-chlorine SOA are below the average value observed for ambient OA influenced by isoprene emission (6.5 ± 2.2 ‰) and much lower than IEPOX-derived SOA (12-40 ‰) observed in laboratory studies (Hu et al., 2015). We also attempted to but were unable to extract an IEPOX factor using positive matrix factorization (Ulbrich et al., 2009), as some studies have done (Budisulistiorini et al., 2013; Lin et al., 2012). Reduction in gas-phase products including those resembling IEPOX was also observed in the CIMS when the aerosol was acidic (see Fig. S14). These observations are consistent with increased partitioning of gas-phase products to the aerosol when the seed aerosol is acidic, resulting in the higher SOA concentrations shown in Fig. 1 and Table 1, but do not prove the presence of IEPOX-derived SOA. Nevertheless, it is clear that SOA formation from chlorine-initiated and OH-initiated oxidation of isoprene proceeds via multi-generational oxidation chemistries involving similar, if not identical, key gas-phase products such as MACR and multifunctional hydroperoxide compounds.

**4 Conclusions**

Chlorine-initiated oxidation of isoprene under low $NO_x$ was investigated inside an environmental chamber. Chlorine was injected either in bulk or continuously in low amounts to simulate fast and slower oxidation conditions. Prompt SOA formation was observed in both cases, indicative of the initial formation of low volatility products. The average wall loss-corrected SOA yield was 8 ± 1 % for continuous chlorine injection experiments and 20 ± 3 % for initial injection experiments; the differences likely resulted from a combination of organic vapor wall loss, oxidative fragmentation, and photolysis of SOA. For air quality models which do not explicitly account for SOA aging, the averaged SOA yield from continuous chlorine injection experiments (8%) should be used for SOA formation from chlorine-initiated oxidation of isoprene. The presence of acidified seed aerosol is shown to enhance SOA formation under low RH conditions. The extent of oxidation of SOA from chlorine-initiated oxidation of isoprene was similar to that of SOA formed from OH-oxidation of isoprene under high $NO_x$ conditions. CIMS measurements identified several initial gas-phase oxidation products as potential SOA precursors including $C_5H_6O$ (MBO), $C_5H_8O$ (2-methylbut-3-enal), $C_5H_7ClO$ (CMBO) and $C_4H_6O$ (MVK/MACR). Gas-phase data provide evidence of chlorine-initiated secondary OH chemistry and its potential contribution to SOA formation. Ions consistent with hydroperoxides and chloroalkyl hydroperoxides were observed in the CIMS and corresponding $HO_x$-enabled formation pathways are proposed.

SOA formation from chlorine-initiated oxidation of isoprene is reported for the first time. The high isoprene-chlorine SOA yields suggest that, despite comparatively low ambient abundance, chlorine radicals could have a notable contribution to overall SOA formation. Proposed formation pathways and gas-phase measurements by the CIMS show that chlorine-initiated oxidation of isoprene could produce chloroalkyl hydroperoxide species, analogous to the formation of low-volatility

hydroperoxides observed for OH-isoprene oxidation under low $NO_x$ conditions. These may contribute to the high observed yields, as well as the relatively fast reduction of organic aerosol mass under UV. Challenges associated with the detection and quantification of particulate organochlorides suggest that the prevalence of ambient particulate chlorine is likely underestimated. Overall, our findings indicate that tropospheric chlorine chemistry increases the oxidative capacity of the atmosphere and directly contributes to SOA formation.

## Acknowledgements

This material is based upon work supported by the Welch Foundation under Grant No. F-1925 and the National Science Foundation under Grant No. 1653625. The work was also funded in part with funds from the State of Texas as part of the program of the Texas Air Research Center. The contents do not necessarily reflect the views and policies of the sponsor nor does the mention of trade names or commercial products constitute endorsement or recommendation for use.

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

**Table 1.** Summary of experimental conditions and results

| Exp.[a] | Isoprene (ppb)[b] | Chlorine (ppb)[c] | VOC:Cl$_2$ | Cl$_2$ Inj.[d] | Ini Seed SA (um$^2$ cm$^{-3}$)[e] | Seed Acidity | Max OA (µg m$^{-3}$)[f] | Yield |
|---|---|---|---|---|---|---|---|---|
| A1 | 24 | 40 | 0.60 | Ini | 1500 | Neutral | 12 | 0.18 |
| A2 | 40 | 88 | 0.45 | Ini | 900 | Neutral | 23 | 0.21 |
| A3 | 82 | 120 | 0.68 | Ini | 810 | Neutral | 39 | 0.17 |
| A4 | 120 | 180 | 0.67 | Ini | 1300 | Neutral | 79 | 0.24 |
| A5 | 160 | 230 | 0.70 | Ini | 1800 | Neutral | 88 | 0.20 |
| A6 | 98 | 100 | 0.98 | Ini | 3700 | Acidic | 80 | 0.29 |
| A7 | 98 | 100 | 0.98 | Ini | 2700 | Acidic | 70 | 0.26 |
| A8[h] | 180 | 360 | 0.50 | Ini | 780 | Neutral | 180 | 0.36 |
| | | | | | | | | |
| C1 | 40 | 94 | 0.42 | Cont | 950 | Neutral | 9 | 0.08 |
| C2 | 120 | 180 | 0.67 | Cont | 880 | Neutral | 32 | 0.10 |
| C3 | 240 | 270 | 0.89 | Cont | 770 | Neutral | 53 | 0.08 |
| C4 | 300 | 250 | 1.20 | Cont | 1400 | Neutral | 60 | 0.07 |
| C5[g] | 73 | 120 | 0.61 | Cont | N/A | Neutral | N/A | N/A |
| C6 | 98 | 180 | 0.54 | Cont | 2400 | Acidic | 41 | 0.15 |
| C7 | 120 | 150 | 0.80 | Cont | 3200 | Acidic | 70 | 0.21 |

[a] "A" for initial Cl$_2$ injection experiments and "C" for continuous Cl$_2$ injection experiments.

[b] Initial isoprene concentration.

[c] Amount of chlorine injected initially (for A1–A8) or cumulatively (for C1–C7).

[d] Chlorine injection method: "Ini" for initial injection; "Cont" for continuous injection.

[e] Total initial surface area of ammonium sulfate or acidified seed aerosol.

[f] Highest, particle wall-loss-corrected OA concentration observed.

[g] No SEMS data available.

[h] ACSM scan speed for A8 (500 ms amu$^{-1}$) was different from other experiments (200 ms amu$^{-1}$).

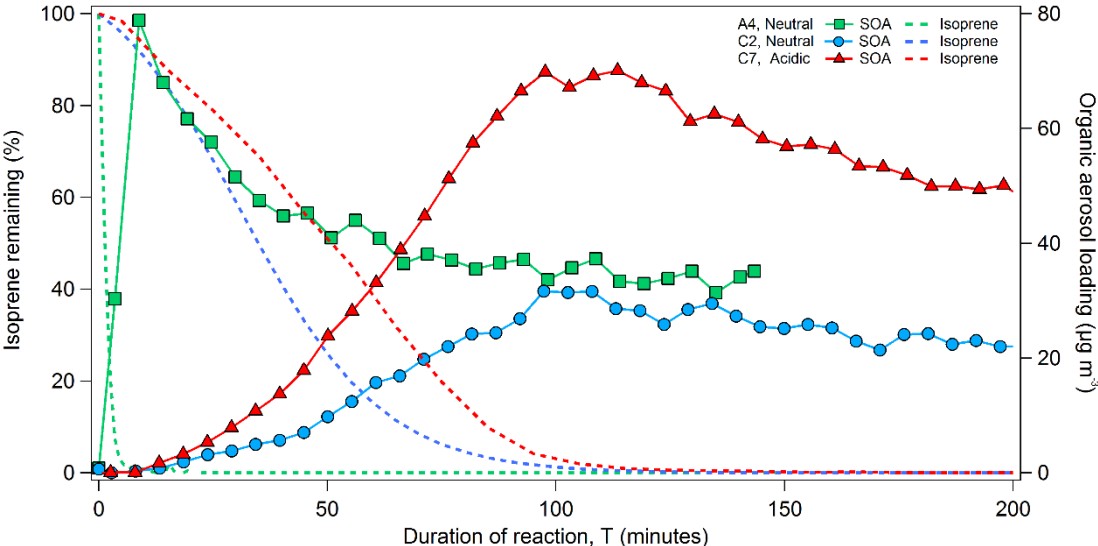

**Figure 1:** Comparison of SOA formation during continuous (C2, C7) and initial (A4) $Cl_2$ injection experiments with similar precursor concentrations but different seed aerosol acidities (C2 vs. C7). SOA concentration is wall-loss corrected and averaged over five-minute intervals. Isoprene concentration is tracked using $(H_2O)_nH_3O^+$ CIMS and averaged over one-minute intervals. Maximum SOA concentrations are reached when isoprene has been depleted. After extended oxidation, observed OA loadings are similar for initial chlorine and continuous chlorine experiments using neutral seed particles. Peak SOA concentration reached in the presence of acidic aerosol during Exp. C7 is much higher than peak SOA concentration reached using neutral seed during Exp. C2.

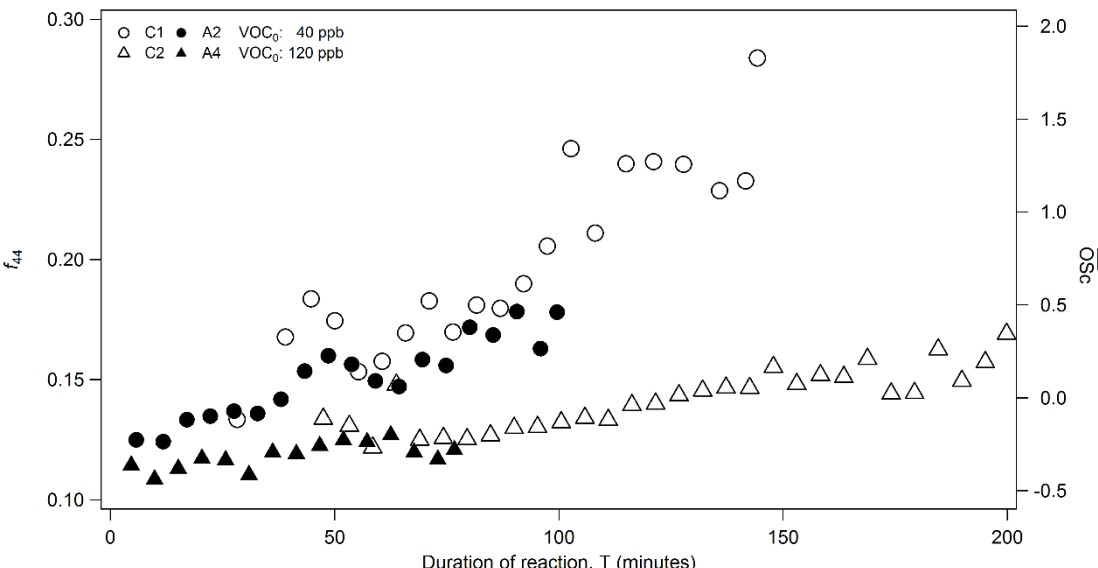

**Figure 2:** Comparison of SOA aging between two pairs (C1 and A2; C2 and A4) of initial and continuous $Cl_2$ injection experiments. The trend of $f_{44}$ is consistent for each pair, regardless of chlorine injection method used. Higher initial isoprene concentration resulted in less oxidized organic aerosol (lower $f_{44}$). $\overline{OS}_C$ is estimated based on $f_{44}$ (see section S2).

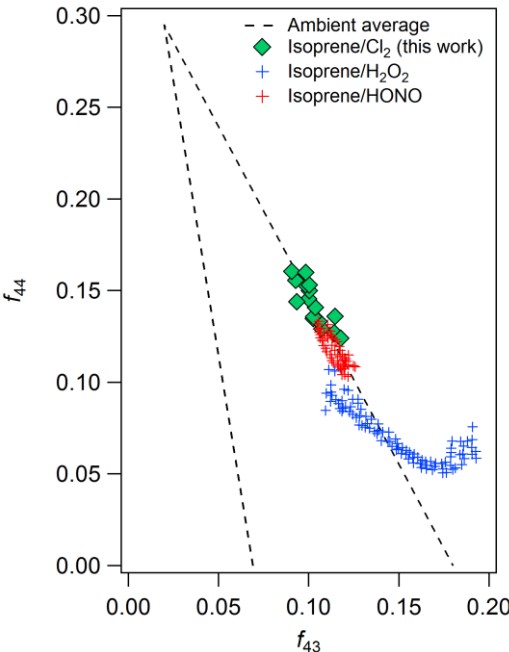

**Figure 3:** Extent of oxidation of SOA generated from chlorine-initiated reactions (Exp. C3, five-minute averages) compared to OH-oxidation of isoprene under low- and high-NO$_x$ (Chhabra et al., 2011). Area enclosed by the dashed lines represent typical ambient OA measurements (Ng et al., 2010).

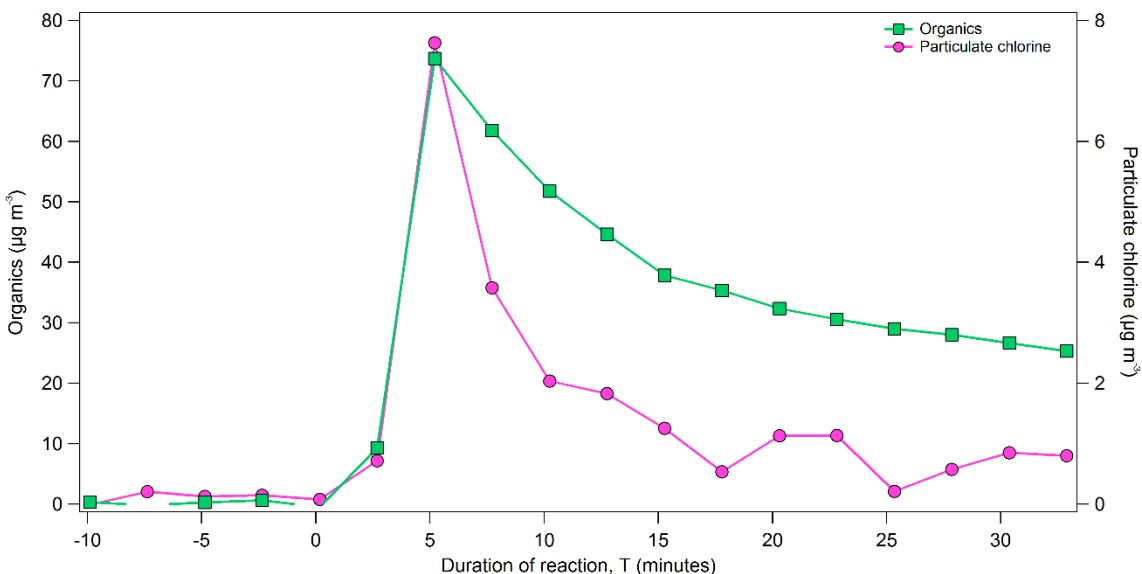

**Figure 4:** Measurements of particulate organics and chlorine from Exp. A8, not corrected for wall losses. UV was turned on at T = 0 mins. Apparent particulate chlorine concentration rapidly decreased to near-background levels while significant suspended OA mass remained. This rapid decrease is likely a measurement artifact due to build-up of chlorinated compounds on the vaporizer surface, see section 3.2 and S4.

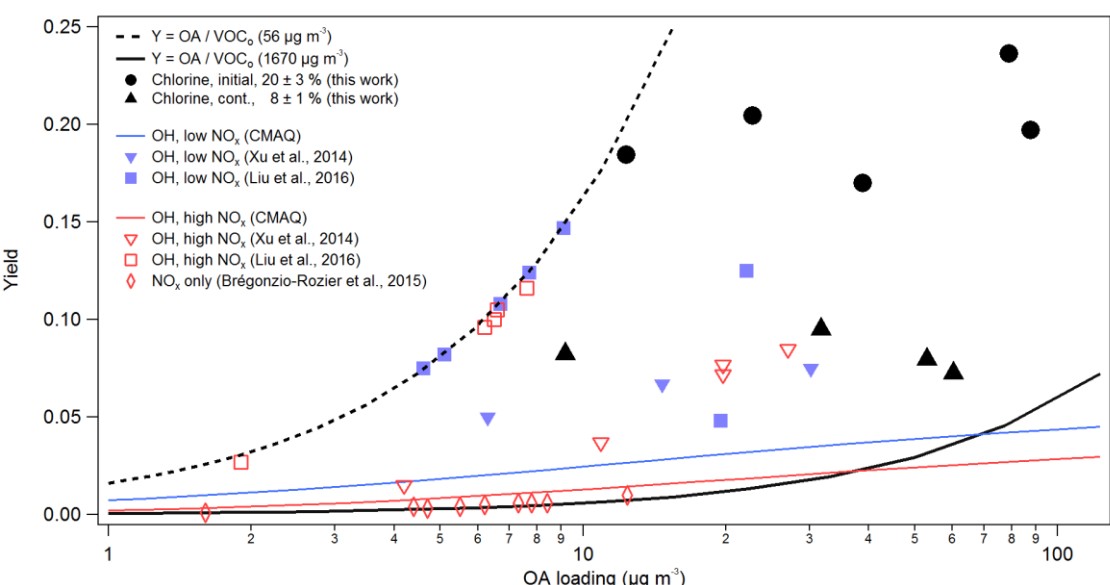

**Figure 5:** Comparison of observed isoprene-chlorine SOA yield with recent literature values corresponding to low and high $NO_x$ OH-oxidation. Dashed lines illustrate the concept of a "pre-defined" yield curve as discussed in the text. The yield used in CMAQ was reproduced using cited VBS parameters (Koo et al., 2014).

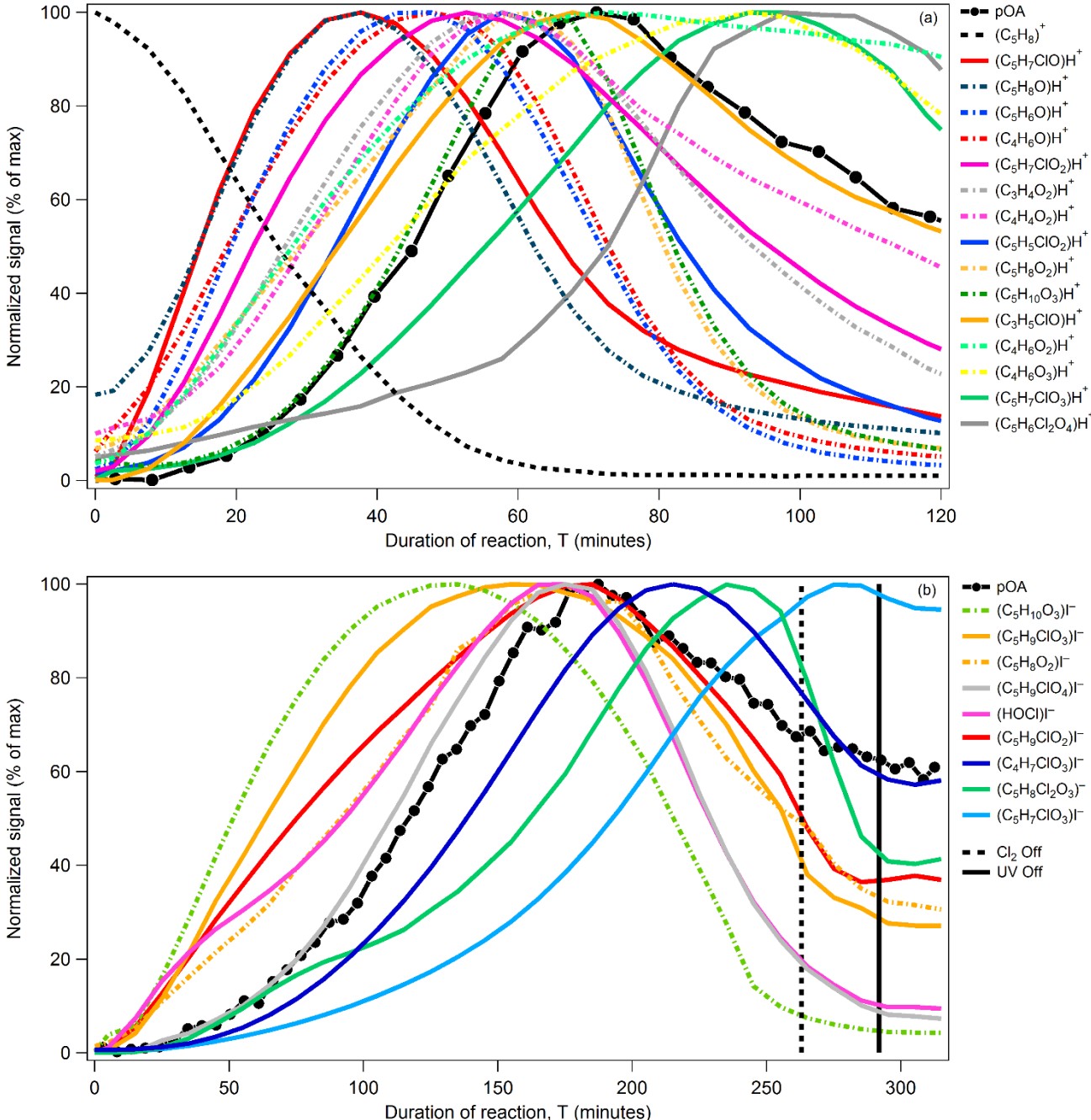

**Figure 6:** Observation of gas-phase species (a) using $(H_2O)_nH_3O^+$ CIMS from Exp. C5. and (b) using $(H_2O)_nI^-$ CIMS from Exp. C3. Suspended particle concentration from each experiment is shown. Ion signal for each individual species was normalized to its maximum value. Five-minute averages are shown. Chlorine injection stopped at T = 263 mins and UV lights were turned off at T = 292 mins during Exp. C3. Species displayed consist of previously reported products in literatures and multigenerational products proposed in Figure 7. $(C_5H_{10}O_3)I^-$ potentially represent ISOPOOH or IEPOX, while $(C_5H_{10}O_3)H^+$ is likely $(C_5H_8O_2)(H_3O)^+$, an oxidation product. $(C_4H_7ClO_3)I^-$ may also represent $(C_4H_5ClO_2)(H_2OI)^-$.

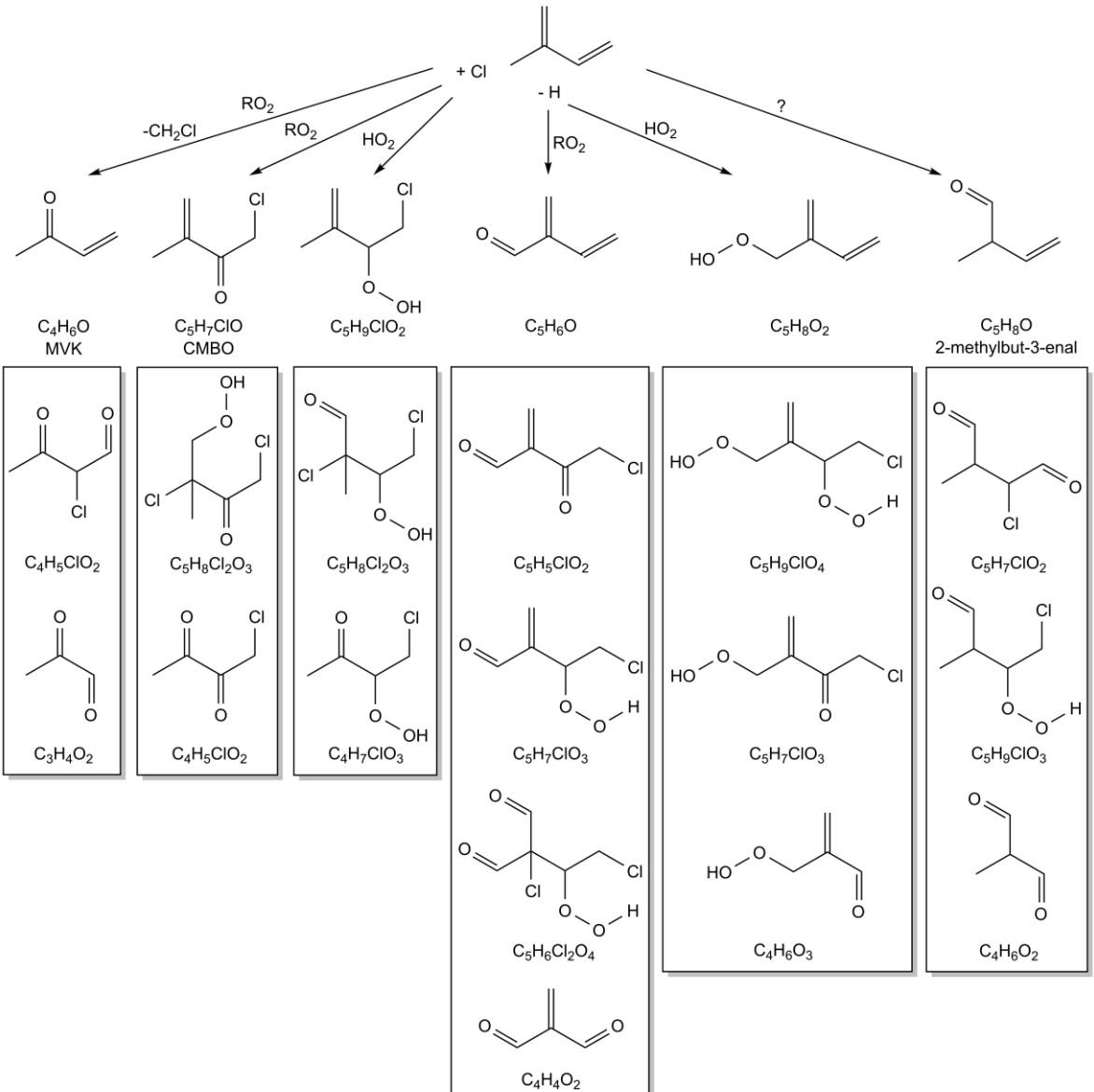

**Figure 7:** Select oxidation products formed under NO$_x$-free conditions. More detailed individual reaction pathways are shown in Fig S10-12. For illustrative purposes, only one isomer is shown per reaction pathway type (e.g. H-abstraction vs. Cl-addition; RO$_2$ + RO$_2$ vs. RO$_2$ + HO$_2$; C-C cleavage vs. no cleavage). As explained in the text, HO$_2$ radicals required for certain reaction pathways were produced from preceding RO$_2$ + RO$_2$ chemistry; formation mechanism for C$_5$H$_8$O is unclear. Time series of all product shown can be found in either Fig 6a or 6b.