# Peer review of "Secondary organic aerosol from chlorine-initiated oxidation of isoprene"

_Atmospheric Chemistry and Physics, 2017_

## Referee Comment (RC1) · Anonymous Referee #1 · 5 Jun 2017

Wang and Hildebrandt Ruiz present a laboratory study of secondary organic aerosol (SOA) formation from chlorine-initiated oxidation of isoprene in chamber experiments under low relative humiditiy and low NOx conditions. They report maximum SOA yields from two different types of experiments, the initial injection of chlorine, and the continuous injection of chlorine. In both cases, prompt SOA formation was observed. In the light of recent observations of unexpectedly high reactive chlorine concentrations in the atmosphere, this study contributes important findings for a better evaluation of the role of chlorine-initiated oxidation reactions of organic compounds in the atmosphere. The supplement gives comprehensive additional information, e.g. a valuable discussion of quantification issues of particulate chloride with the ACSM/AMS. In my opinion, the

study is highly topical and deserves publication in ACP after taking into account the following comments:

a) p.5, line 1: The limitations of estimating O:C, H:C, and the oxidation state of carbon from f44 based on empirical correlations should be briefly discussed. For example, equation S-1 in section S2 may underestimate O:C values substantially in environments dominated by NOx-free isoprene chemistry (Canagaratna et al., 2015). Also, the presence of heteroatoms may introduce deviations from equation S-3 in section S2 when estimating the oxidation state of carbon (Kroll et al., 2011).

b) p.5, line 13: The separation of experiments A1-A5 and experiments H1/H2 seems to be somewhat arbitrary. In my opinion, experiment H1 should be experiment A6, and the much higher maximum yield of this experiment should be part of the discussion of section 3.3. Experiment H2 is a technical experiment to "...explore the ability of the ACSM to detect organic chloride" (supplement, section S4). I was confused to find information about H2 in Table 1, and I recommend to remove it from the table and just explain the character of this experiment in the supplement.

c) p.5, line 21: Chlorinated organic compounds have also been identified in ambient aerosol samples from Western Australia by ion cyclotron mass spectrometry (Kamilli et al., 2016), with a higher abundance of chlorinated organic compounds in daytime samples when photochemistry is active.

d) p.7, line 19: The VOC:Cl2 ratios may be expected to be much higher under atmospheric conditions than in the presented experiments. Do the authors have some insight, or could they speculate about how the yields may change for larger isoprene:Cl2 ratios? Also, when presenting the highest observed SOA yields, why do the authors exclude experiment A1 for the average yield of the inital injection experiments?

e) p.8, line 8: When discussing secondary OH chemistry, the authors mention potentially unidentified HOx production pathways other than HO2 production during formation of CMBO. It would be extremely interesting to have at least a semi-quantitative

estimate of the contributions of chlorine-initiated secondary OH chemistry vs. OH chemistry from other sources, potentially also due to chamber wall effects.

Technical comments:

in manuscript: p.2, line 16 and p.7, line.27: When referring to isomers of CMBO, these should be isomers of chloromethylbutenone, e.g. 1-chloro-3-methyl-3-butene-2-one, not "isomers of 3-methyl-3-butene-2-one". p.3, line 23: Change "relatively ionization efficiencies" to "relative ionization efficiencies". p.4, line 10: The reference should read "Odum et al., 1996". p.9, line 4: Change "produced form" to "produced from". p.9, line 6: Change "chlorine-initiation oxidation" to "chlorine-initiated oxidation". p.18, Table 1: I don't understand the value of the VOC:Cl2 ratio in experiment H2.

in supplement: p.3, line 7: Change "number of water cluster" to "number of water clusters". p.4, line 7: Remove "greater than" before "44 % overestimation could be expected".

References

Canagaratna, M. R., Jimenez, J. L., Kroll, J. H., Chen, Q., Kessler, S. H., Massoli, P., Hildebrandt Ruiz, L., Fortner, E., Williams, L. R., Wilson, K. R., Surratt, J. D., Donahue, N. M., Jayne, J. T. and Worsnop, D. R.: Elemental ratio measurements of organic compounds using aerosol mass spectrometry: Characterization, improved calibration, and implications, Atmos. Chem. Phys., 15, 253-272, doi:10.5194/acp-15-253-2015, 2015.

Kamilli, K.A., Ofner, J., Krause, T., Sattler, T., Schmitt-Kopplin, P., Eitenberger, E., Friedbacher, G., Lendl, B., Lohninger, H., Schöler, H., Held, A.: How salt lakes affect atmospheric new particle formation: A case study in Western Australia. Sci. Total Environment, 573, 985-995, 2016.

Kroll, J. H., Donahue, N. M., Jimenez, J. L., Kessler, S. H., Canagaratna, M. R., Wilson, K. R., Altieri, K. E., Mazzoleni, L. R., Wozniak, A. S., Bluhm, H., Mysak, E. R.,

Smith, J. D., Kolb, C. E. and Worsnop, D. R.: Carbon oxidation state as a metric for describing the chemistry of atmospheric organic aerosol., Nat. Chem., 3, 133-139, doi:10.1038/nchem.948, 2011.

---

## Referee Comment (RC2) · Anonymous Referee #3 · 7 Jun 2017

The manuscript "Secondary organic aerosol from chlorine-initiated oxidation of isoprene" by Wang and Ruiz reports SOA yields ranging from 8 to 36% derived from chlorine-initiated isoprene oxidation. These yields are obtained by two different types of aerosol smog-chamber experiments, continuous and initial chlorine injection. The presented work is of highest scientific quality and the manuscript is well written. I therefor recommend publication in ACP after concerning the following minor comments:

Introduction: The authors should add a Paragraph to the introduction about natural and anthropogenic halogen sources and sinks in the atmosphere to introduce this topic to the readers; e.g. by: Simpson et al., Tropospheric Halogen Chemistry: Sources,

[Figure]

Cycling, and Impacts, Chem. Reviews, 2015. Roland von Glasow, Wider role for airborne chlorine, nature, 464, 2010. Finlayson-Pitts, Halogens in the Troposphere, Anal. Chem., 82, 770-776, 2010. Buxmann et al., Consumption of reactive halogen species from sea-salt aerosol by secondary organic aerosol: slowing down the bromine explosion, Environ. Chem., 12, 476-488, 2015.

p2 line 30: Please add the characteristics of the UVA light source: actinic flux, quantified UV/VIS spectrum.

P3 line 31 "loss of organic vapors to Teflon surfaces" Teflon films, used for aerosol smog-chambers, are known to store various gaseous species, especially NOx, which is released from the Teflon film by UV radiation and increased temperatures. Has this been observed or taken into account? Please add a related statement to the manuscript.

---

## Referee Comment (RC3) · Anonymous Referee #4 · 10 Jun 2017

Summary and Recommendation:

This is an interesting study that examines the potential of SOA formation from chlorine radical (Cl)-initiated oxidation of isoprene. I've spent many years examining the detailed chemical processes of isoprene oxidation leading to SOA formation, and to my knowledge, this does appear to be the first examination of the role of Cl radicals in forming SOA. The paper is concise and will be of great interest to the readers of ACP. I'm particularly interested in this work as my group has recently been examining the potential of isoprene SOA in marine aerosols, especially from remote regions where NOx levels are likely low. As one of the reviewers recommended in the quick review

stage, I think it is important to more clearly stress where this chemistry might be relevant that you are examining in here in your chamber studies. I stress this point as we definitely measure isoprene SOA tracers related to low-NOx pathways in remote marine aerosols (work yet to be published). Field studies have also demonstrated this, such as Fu et al. (JGR, 2011) and Hu et al. (Scientific Reports, 2013). However, one thing that is interesting to us is could other oxidants play a role. This study seems to suggest this!

I have a few major points that need to be addressed before publication can be fully considered. Due to the nature of these major points, I think the manuscript should be accepted with major revisions noted. Once the authors can address these, I will certainly support publication in ACP. Briefly, I'll summarize that I think the chamber experiments could have included other conditions (like higher RH or more acidic aerosol) to examine SOA potential from isoprene oxidation, more references should have been cited, and importantly I think that PMF/ME-2 analyses of the ACSM data set should have been considered in order to estimate if IEPOX-derived SOA or non-IEPOX SOA dominated to the SOA mass. This sort of mass closure would reveal or provide more context on what other potential pathways/products are missing (such as organochlorine products).

Major Points to Consider:

1.) As the authors know well, acidity plays a MAJOR role for IEPOX uptake yielding isoprene SOA under low-NOx conditions. This was conclusively demonstrated with authentic IEPOX for the first time by Lin et al. (2012, ES&T); however, Wang et al. (2005, RCM), Surratt et a. (2006, JPCA), Paulot et al. (2009, Science), and Surratt et al. (2010, PNAS) were some of the first studies to propose for the existence of IEPOX even though an authentic standard did not exist at that time to study its reactive uptake. Since then, kinetic studies have demonstrated that acidity plays a key role in IEPOX producing substantial amounts of SOA (Gaston et al., 2014, ES&T; Riedel et al., 2015, ES&T Letters; Riedel et al., ACP, 2016). If ammonium sulfate aerosol is wet,

due to a high enough RH, then ammonium sulfate can take up IEPOX to yield SOA if the reaction time scales are long enough (Nguyen et al., 2014, ACP).

With this reminder above, I wonder why the authors did not consider also conducting experiments at elevated RH and increased acidities with the ammonium sulfate seed aerosol? I can imagine if the chemistry applies to remote marine locations, the aerosol may be more wet and/or acidic (especially if there are sufficient DMS emissions). Jon Abbatt's group also showed recently in Wong et al. (2015, ES&T) that deliquesced ammonium sulfate particles can yield a lot of SOA through a non-IEPOX route. So this could be something important to consider.

By the way, the authors don't appear to say how the ammonium sulfate aerosol were injected into the chamber? What was the concentration of your atomizing solution? This should be added to the experimental section.

2.) PMF/ME-2 analyses of your SOA composition using the ACSM data:

As the authors likely know, Lin et al. (2012, ES&T), and more specifically Budisulistiorini et al. (2013, ES&T), demonstrated that AMS and ACSM, respectively, datasets can resolve IEPOX-OA factor when PMF is applied. Why didn't the authors consider conducting PMF in their analyses to constrain how much of the SOA is from IEPOX? You could run PMF with the IEPOX-OA factor constrained using the reference MS library (so this would be ME-2).

Furthermore, Krechmer et al. (2015, ES&T) did this for the non-IEPOX SOA pathway. He used his reference mass spectrum for the non-IEPOX SOA to constrain its importance to field aerosol collected during the 2013 SOAS campaign! Riva et al. (2016, ES&T) also showed that authentic ISOPOOH makes SOA without needing IEPOX due to the low-volatility nature of the multifunctional hydroperoxides produced!

Since you don't use offline chemical analyses to measure molecular-level SOA components, I think it is worth while conducting PMF/ME-2 analyses to see if that can help

constrain the different pathways yielding the total SOA mass. I hope the authors might agree with this suggestion.

3.) Please go through carefully and make sure certain references are not missing throughout the text. I mention a few of these in my minor comments below.

4.) I know it isn't a focus in this manuscript, but it would be very powerful if molecular tracers could be identified for Cl-initiated radicals yielding SOA. The authors mention using the ACSM to try to constrain the organochlorine budget, but seemed to have trouble with this due to interference issues. This is why I suggested conducting PMF/ME-2 analyses above in # 2. However, does the CIMS data (especially the iodide reagent ion chemistry) suggest the presence of low-volatility hydroperoxides that contain chlorine in them? From OH radical studies by Krechmer et al. (2015, ES&T), Riva et al. (2016, ES&T), and Liu et al. (2016, ES&T), they all measured low-volatility multifunctional hydroperoxides that made sufficient amounts of SOA (that don't require aerosol acidity like IEPOX).

5.) When reviewing Table 1, I realized it wasn't well explained in the text why the different injection methods were used. What did these methods explicitly tell you? For modelers, this Table might be very difficult for them to judge which yields should be used. Also, related to my point # 1 above, modelers seeing these yields may question if these yields are accurate to remote low-NOx regions where Cl radical chemistry might matter. Can the authors offer which yield may be the most appropriate to use?

Finally, I'm assuming these various injection methods were used to gain some insights into vapor wall losses? It remains unclear to me how exactly vapor wall losses were dealt with (if at all) in reporting the SOA yields shown in Table 1.

Minor Comments:

1.) Abstract, Page 1, Line 12: Remove "%" after "8." You don't need this.

2.) Methods, Page 2, Line 32: Insert a space between "exceeding100"

3.) Section 3.4, Page 7, Line 27: You write "3-methyl-3-butene-2-one (CMBO) [(C5H7OCl)H+]." This appears to be named incorrectly. Please name according to IUPAC.

4.) Section 3.4, Page 7, Line 31: I would reference Kroll et al. (2006, ES&T) and Surratt et al. (2006, JPCA) as one of the initial references to demonstrates MACR oxidation is a source of SOA.

5.) Section 3.4, Page 8, Line 26: The authors should reference Lin et al. (2012, ES&T)

6.) Section 3.4, Page 8, Line 27: The authors should also reference Gaston et al. (2014, ES&T) and Riedel et al. (2015, ES&T Letters)

---

## Referee Comment (RC4) · Anonymous Referee #2 · 12 Jun 2017

Wang and Hildebrandt Ruiz provide the results of chamber studies of the low NOx chlorine oxidation of isoprene, the most abundantly-emitted BVOC globally. The most valuable aspects of the work are the determination of SOA yields and application of ACSM and CIMS measurements to the study. The application of these new techniques provide a significant opportunity for an in-depth/detailed study of the SOA formation. I encourage the authors to delve further into the mechanistic details of the isoprene-Cl-SOA production, perhaps through comparison of ACSM mass spectra between experiments (or during an experiment) and more in-depth comparison of the gas-phase data with previous isoprene-Cl oxidation studies reported in the literature, as these activities may provide new insights. It may be useful to construct a schematic of possible

reaction/oxidation pathways with highlights of key molecules, particularly since the authors suggest that the major reaction products are similar to OH oxidation. At the very least, it would be helpful to discuss potential oxidation pathways further, by considering temporal trends in the various gas-phase species that suggest different generations of trace gases. Perhaps you could calculate the O:C ratios of these molecules and estimate associated vapor pressures for support and to consider partitioning to the particle phase. Overall, it seems like more information can likely be extracted from the ACSM and CIMS data.

Major Comments:

- Introduction: In motivating their laboratory study, the authors may wish to point out marine emissions of isoprene and the consideration of isoprene SOA from this source (e.g. Gantt et al 2010, Atmos. Environ.). The authors currently do not consider the potential for Cl oxidation of isoprene far from coasts (where NOx may also be low), which is a motivating factor for this work.

- Throughout the manuscript, chlorine incorporated into organic molecules appears to be referred to as "chloride", which chemically refers to Cl-, rather than chloro-organics, or organic chlorine. This reference to chloride is confusing as it makes the reader question whether the authors are indeed suggesting that inorganic chloride is present in the particle phase. This needs to be clarified throughout.

- Sec 2.2, Instrumentation, should be clarified in the main text in terms of the description of the CIMS, for which additional information is needed. Perhaps material from the supplemental information should be moved here, in addition to revisions for clarity. It is stated that proton transfer, charge exchange, and clustering are all used for chemical ionization, which is confusing since typically one pathway is chosen through specific conditions within the ion molecule region of the instrument. As worded, it sounds like these reaction pathways of analyte ion formation are all occurring simultaneously. It is also odd to me that the instrument doesn't seem to have been tuned for conditions of

primarily H3O+, rather than (H2O)nH3O+. What fraction of the signal was associated with H3O+, and how many n were observed? This would impact the resulting analyte ionization. When were (H2O)nH3O+ vs (H2O)nI- reagent ions used? Did this switch back and forth during experiments, or was one ion chemistry used per experiment? Were CIMS experiments conducted during all experiments, or only during C3 and C5?

- Page 5, First Paragraph: Was a decrease in reagent ion signal observed compared to below an experiment? This might suggest a non-linear response and concern that reagent ion reactions could be limited even if still in excess. Without calibration of the signals, this would make trends more difficult to assess if in a non-linear regime. The phrasing on lines 5-6 about this is not clear. Also, why wasn't at least isoprene calibrated for since each experiment started with a known mole ratio? It seems like that would be beneficial to this work and could probably even be done retroactively with knowledge of the experimental parameters. Was "significant depletion of reagent ion" (Page S3, Line 24) observed during any experiment?

- In the results and discussion, it would often be helpful, when possible, to give values in parentheses, rather than vague descriptors so that you don't require the reader to review and correctly interpret the graphs.

- Section 3.2 should either be moved after section 3.4 or moved to the supplemental information. This section does not contribute much to our understanding of Cl-SOA or precursors, as it primarily focuses on an issue with the ACSM method, which while important, doesn't seem to be the main focus of this work. Rather than only identifying a potential issue, could a chlorinated organic standard be purchased and aerosolized for characterization so that the authors could provide a solution to the problem as well? Similarly, nearly a full paragraph in the conclusions is dedicated to this subject, which detracts from the exciting science studied. Also, use of m/z 36 here is not intuitive when referring to chloride.

- Page 7, Lines 11-12: It is not clear, as written, if you then used a 2D model here.

- Please review rules for significant figures for numbers and fix throughout. Please note that when reporting error only one significant figure should be used, with the same number of decimal places used for the average and the error.

- Can you compare ACSM mass spectra at different points during an experiment to examine possible evidence of oxidative fragmentation or vapor wall loss (as discussed on Page 7, Lines 18-19)? Could ACSM mass spectra be compared between experiments to examine the potential for differences in SOA composition?

- A conclusion of the study is that "The effects of SOA aging must be described explicitly and separately from initial SOA formation." (Page 9, lines 16-17) Yet, few details are given in the results and discussion for what this explicit description is. Above I suggested possible ways to provide greater mechanistic information on the Cl-isoprene oxidation and subsequent SOA formation.

- Another conclusion of the study is "Similarities between chlorine-isoprene and OH-isoprene oxidation products suggest that air quality models may be able to lump the treatment of SOA produced from chlorine- and OH-initiated oxidation of isoprene." Yet this is difficult to discern as very little discussion was dedicated to this important topic. There also appears to be no quantitative information that would indicate similar yields associated with various reaction pathways. More in-depth interpretation and discussion of the data is required to support this statement.

Minor Comments:

- Page 2, Line 11: It is unclear why Riedel et al 2012 is cited here, since that is a coastal marine study.

- Page 4, Line 12: It is confusing to have several equations on the same line. It would be preferable to have one equation per line and number as such.

- Page 4, Lines 17-18: The words "low" and "high" are vague, and it would be useful to include at least approximate values in parentheses as well, for example, to aid in

interpretation of these descriptors.

- Page 6, lines 4-5: This sentence states "high reactivity of chlorine radicals toward isoprene and its reaction products" and therefore seems to contradict the earlier sentence on page 5, lines 27-28.

- Page 6, Line 10: The phrase "quantification proved to be difficult" is vague.

- Page 7, Lines 13-14: I would suggest deleting this sentence, as the previous paragraph already explained this and having this information here as well could confuse the reader.

- Page 7, Line 15: For clarity, I suggest adding the following to the end of the sentence ". . .literature values of OH oxidation under low and high NOx scenarios."

- Page 7, Line 16: Why was the highest observed SOA yield reported, rather than an average, for example?

- Page 7, Line 16: By "continuous cases", do you mean continuous Cl2 injection during an experiment? Make sure this is clear.

- Hypochlorous acid is generally written as HOCl. I'm not used to ClOH, as written throughout.

- Page 9, Lines 1-2: This sentence is commenting on the method, moreso than the science and could be moved to the methods section or supplemental.

- Page 9, Line 16: It isn't clear why this sentence is needed to be highlighted in the conclusions.

- Page 9, Lines 18-24: It would be useful to merge this short paragraph with the first paragraph of the conclusions section.

- Figure 3 caption: This figure does not explicitly show oxidation state as stated in the caption, which is misleading.

- Figure 5 caption: For clarification, I suggest adding the phrase "corresponding to low and high NOx OH oxidation" at the end of the first sentence.

- Figure 6 caption: It is not clear what is meant by "interfering ions" here.

- S1: Please provide references for this section of the SI.

- Page S3, Line 13: Why would instrument sensitivity change over time?

- Page S3, Line 16: k does not appear to be defined.

―――――――――――――――――

---

## Author Comment (AC1) · 16 Aug 2017

We thank the referee for the suggestions and recommendations. Below are our responses to all comments.

(a) Reviewer: p.5, line 1: The limitations of estimating O:C, H:C, and the oxidation state of carbon from f44 based on empirical correlations should be briefly discussed. For example, equation S-1 in section S2 may underestimate O:C values substantially in environments dominated by NOx-free isoprene chemistry (Canagaratna et al., 2015). Also, the presence of heteroatoms may introduce deviations from equation S-3 in section S2 when estimating the oxidation state of carbon (Kroll et al., 2011).

[Figure]

Response: We have added discussion of the limitations of the empirical correlation and ACSM data to the revised manuscript

Manuscript changes in Section 2.3: "The empirical correlations were derived using a comprehensive collection of AMS datasets but may underestimate O:C values for SOA formed under low NOx conditions from isoprene or toluene (Canagaratna et al., 2015). Variability in f43 and f44 among different ACSMs have also been reported. (Crenn et al., 2015)."

Manuscript changes in Section S2: "Deviation from Eq. (S3) could occur due to the presence of peroxide groups or heteroatom groups, such as select chloroalkyl hydroperoxide compounds identified in CIMS measurements (Kroll et al., 2011).

(b) Reviewer: p.5, line 13: The separation of experiments A1-A5 and experiments H1/H2 seems to be somewhat arbitrary. In my opinion, experiment H1 should be experiment A6, and the much higher maximum yield of this experiment should be part of the discussion of section 3.3. Experiment H2 is a technical experiment to "...explore the ability of the ACSM to detect organic chloride" (supplement, section S4). I was confused to find information about H2 in Table 1, and I recommend to remove it from the table and just explain the character of this experiment in the supplement.

Response: We agree with the referee's recommendation. We have re-designated experiment H1 as experiment A8 and have conducted two additional initial chlorine injection experiments which are now designated as A6 and A7. Experiment H2 is now designated as experiment S1. Experimental details for Exp. S1 are now described in-text within supplement section S4.

(c) Reviewer: p.5, line 21: Chlorinated organic compounds have also been identified in ambient aerosol samples from Western Australia by ion cyclotron mass spectrometry (Kamilli et al., 2016), with a higher abundance of chlorinated organic compounds in daytime samples when photochemistry is active.

Response: A reference to this work has been added in section 3.1 of the revised manuscript.

d) Reviewer: p.7, line 19: The VOC:Cl2 ratios may be expected to be much higher under atmospheric conditions than in the presented experiments. Do the authors have some insight, or could they speculate about how the yields may change for larger isoprene:Cl2 ratios? Also, when presenting the highest observed SOA yields, why do the authors exclude experiment A1 for the average yield of the initial injection experiments?

Response: We have added discussion on a potential correlation between the VOC:Cl2 ratio and SOA yields, as well as details that may be of interest to modelers. The yield calculated from experiment A1 was in fact used in calculating the average initial chlorine experiment SOA yields in the discussion paper. The text "A2-A5" was a typographical error and has now been corrected to "A1-A5." Based on new calibration results we have also updated the relative ionization efficiency (RIE) values used in ACSM data analysis, which resulted in lower calculated yield values.

Manuscript changes in Section 3.3: "Under atmospheric conditions, the VOC to chlorine ratio will usually be higher than those used in these experiments. Previous studies on chlorine-initiated SOA formation from toluene (Cai et al., 2008) and limonene (Cai and Griffin, 2006) suggest that SOA yield decreases with higher VOC-to-chlorine ratio. While we do not observe a clear correlation between SOA yield and isoprene-chlorine ratios used in this work (0.49-1.22), such dependence could be present over a wider range of this ratio. For air quality models, the use of the continuous case yield, which is similar to recently reported OH-oxidation yields (Liu et al., 2016; Xu et al., 2014), is more appropriate because the isoprene-to-chlorine ratio is closer to atmospheric conditions. Furthermore, the SOA yields from continuous injection experiments account for fragmentation reactions in the atmosphere (which occur throughout the experiments). Thus, yields form continuous injection experiments should be used in air-quality models which do not explicitly account for fragmentation reactions."

e) Reviewer: p.8, line 8: When discussing secondary OH chemistry, the authors mention potentially unidentified HOx production pathways other than HO2 production during formation of CMBO. It would be extremely interesting to have at least a semi-quantitative estimate of the contributions of chlorine-initiated secondary OH chemistry vs. OH chemistry from other sources, potentially also due to chamber wall effects.

Response: We have carried out some chamber box modeling and expanded the discussion of HOx chemistry. Model results show that chlorine chemistry accounts for the majority of HO2 production when sufficient isoprene is present. Overall, chlorine radicals consume over 99% of the isoprene. Secondary OH-isoprene chemistry has very minor contribution to the overall photooxidation chemistry in these experiments.

Manuscript changes in Section 2.1: "Background effects were estimated using the SAPRC chamber modeling software (http://www.engr.ucr.edu/~carter/SAPRC/) in combination with the Carbon Bond 6 (CB6r2) chemical mechanism which was modified to include basic gas phase inorganic chlorine chemistry in addition to Cl2 and ClNO2 photolysis (Sarwar et al., 2012; Yarwood et al., 2010). Wall effects are represented within the model by a constant emission of nitrous acid (HONO) from the chamber walls on the order of 0.1 ppb min-1, which was determined separately in chamber characterization experiments (Carter et al., 2005)."

Manuscript changes in Section 3.4: "The SAPRC chamber model results indicate that more than 99% of the isoprene reacts with Cl; secondary OH chemistry is therefore only a very minor pathway in these experiments. Model results also show that HO2 production is dominated by isoprene-chlorine chemistry when sufficient isoprene is present, whereas wall effects dominate HO2 production (>60 %) after all isoprene has been consumed. It is worth noting that the model does not explicitly represent Cl-initiated oxidation of reaction products, which can produce additional HOx radicals. Therefore, we expect the actual secondary OH chemistry to be more important than the current model estimation."

Technical Comments and Responses

-Reviewer: in manuscript: p.2, line 16 and p.7, line.27: When referring to isomers of CMBO, these should be isomers of chloromethylbutenone, e.g. 1-chloro-3-methyl-3-butene-2-one, not "isomers of 3-methyl-3-butene-2-one".

Response: We have corrected the naming

-Reviewer: p.3, line 23: Change "relatively ionization efficiencies" to "relative ionization efficiencies".

Response: We have corrected the typographical error.

-Reviewer: p.4, line 10: The reference should read "Odum et al., 1996".

Response: We have corrected the reference.

-Reviewer: p.9, line 4: Change "produced form" to "produced from". p.9, line 6: Change "chlorine-initiation oxidation" to "chlorine-initiated oxidation".

Response: We have corrected the typographical errors

-Reviewer: p.18, Table 1: I don't understand the value of the VOC:Cl2 ratio in experiment H2.

Response: Wrong precursor concentrations were reported for Exp. H2. We have corrected the errors in the revised manuscript.

-Reviewer: p.3, line 7: Change "number of water cluster" to "number of water clusters". p.4, line 7: Remove "greater than" before "44 % overestimation could be expected".

Response: We have modified the texts as suggested.

References

Cai, X. and Griffin, R. J.: Secondary aerosol formation from the oxidation of biogenic hydrocarbons by chlorine atoms, J. Geophys. Res. Atmos., 111(14), 7348–7359,

doi:10.1029/2005JD006857, 2006.

Cai, X., Ziemba, L. D. and Griffin, R. J.: Secondary aerosol formation from the oxidation of toluene by chlorine atoms, Atmos. Environ., 42(32), 7348–7359, doi:10.1016/j.atmosenv.2008.07.014, 2008.

Canagaratna, M. R., Jimenez, J. L., Kroll, J. H., Chen, Q., Kessler, S. H., Massoli, P., Hildebrandt Ruiz, L., Fortner, E., Williams, L. R., Wilson, K. R., Surratt, J. D., Donahue, N. M., Jayne, J. T. and Worsnop, D. R.: Elemental ratio measurements of organic compounds using aerosol mass spectrometry: Characterization, improved calibration, and implications, Atmos. Chem. Phys., 15(1), 253–272, doi:10.5194/acp-15-253-2015, 2015.

Carter, W. P. L., Cocker, D. R., Fitz, D. R., Malkina, I. L., Bumiller, K., Sauer, C. G., Pisano, J. T., Bufalino, C. and Song, C.: A new environmental chamber for evaluation of gas-phase chemical mechanisms and secondary aerosol formation, Atmos. Environ., 39, 7768–7788, 2005.

Crenn, V., Sciare, J., Croteau, P. L., Verlhac, S., Fröhlich, R., Belis, C. A., Aas, W., Äijälä, M., Alastuey, A., Artiñano, B., Baisnée, D., Bonnaire, N., Bressi, M., Cana-garatna, M., Canonaco, F., Carbone, C., Cavalli, F., Coz, E., Cubison, M. J., Esser-Gietl, J. K., Green, D. C., Gros, V., Heikkinen, L., Herrmann, H., Lunder, C., Minguillón, M. C., Močnik, G., O'Dowd, C. D., Ovadnevaite, J., Petit, J. E., Petralia, E., Poulain, L., Priestman, M., Riffault, V., Ripoll, A., Sarda-Estève, R., Slowik, J. G., Setyan, A., Wiedensohler, A., Baltensperger, U., Prévôt, A. S. H., Jayne, J. T. and Favez, O.: AC-TRIS ACSM intercomparison - Part 1: Reproducibility of concentration and fragment re-sults from 13 individual Quadrupole Aerosol Chemical Speciation Monitors (Q-ACSM) and consistency with co-located instruments, Atmos. Meas. Tech., 8(12), 5063–5087, doi:10.5194/amt-8-5063-2015, 2015.

Kamilli, K.A., Ofner, J., Krause, T., Sattler, T., Schmitt-Kopplin, P., Eitenberger, E., Friedbacher, G., Lendl, B., Lohninger, H., Schöler, H., Held, A.: How salt lakes affect

atmospheric new particle formation: A case study in Western Australia. Sci. Total Environment, 573, 985-995, 2016.

Kroll, J. H., Donahue, N. M., Jimenez, J. L., Kessler, S. H., Canagaratna, M. R., Wilson, K. R., Altieri, K. E., Mazzoleni, L. R., Wozniak, A. S., Bluhm, H., Mysak, E. R., Smith, J. D., Kolb, C. E. and Worsnop, D. R.: Carbon oxidation state as a metric for describing the chemistry of atmospheric organic aerosol., Nat. Chem., 3(2), 133–139, doi:10.1038/nchem.948, 2011.

Liu, J., D'Ambro, E. L., Lee, B. H., Lopez-Hilfiker, F. D., Zaveri, R. A., Rivera-Rios, J. C., Keutsch, F. N., Iyer, S., Kurten, T., Zhang, Z., Gold, A., Surratt, J. D., Shilling, J. E. and Thornton, J. A.: Efficient Isoprene Secondary Organic Aerosol Formation from a Non-IEPOX Pathway, Environ. Sci. Technol., 50(18), 9872–9880, doi:10.1021/acs.est.6b01872, 2016.

Sarwar, G., Simon, H., Bhave, P. and Yarwood, G.: Examining the impact of heterogeneous nitryl chloride production on air quality across the United States, Atmos. Chem. Phys., 12(14), 6455–6473, doi:10.5194/acp-12-6455-2012, 2012.

Xu, L., Kollman, M. S., Song, C., Shilling, J. E. and Ng, N. L.: Effects of NOx on the volatility of secondary organic aerosol from isoprene photooxidation, Environ. Sci. Technol., 48(4), 2253–2262, doi:10.1021/es404842g, 2014.

Yarwood, G., Jung, J., Whitten, G. Z., Heo, G., Mellberg, J. and Estes, M.: Updates to the Carbon Bond Mechanism for Version 6 (CB6), in Presented at the 9th Annual CMAS Conference, Chapel Hill, NC, October 11-13, vol. 6, pp. 1–4., 2010.

---

## Author Comment (AC2) · 17 Aug 2017

We thank the referee for the suggestions and recommendations. Below are our responses to the comments.

(a) Reviewer: Introduction: The authors should add a Paragraph to the introduction about natural and anthropogenic halogen sources and sinks in the atmosphere to introduce this topic to the readers; e.g. by: Simpson et al., Tropospheric Halogen Chemistry: Sources, Cycling, and Impacts, Chem. Reviews, 2015. Roland von Glasow, Wider role for airborne chlorine, nature, 464, 2010. Finlayson-Pitts, Halogens in the Troposphere, Anal. Chem., 82, 770-776, 2010. Buxmann et al., Consumption of re-

[Figure]

active halogen species from sea-salt aerosol by secondary organic aerosol: slowing down the bromine explosion, Environ. Chem., 12, 476-488, 2015.

Response: We have added a discussion on natural and anthropogenic halogen sources and sinks to the introduction of the revised manuscript as suggested by the reviewer.

(b) Reviewer: p2 line 30: Please add the characteristics of the UVA light source: actinic flux, quantified UV/VIS spectrum.

Response: We have included additional information on the UV light source.

Manuscript changes in Section 2.1: "The UV spectrum is similar to other blacklight sources reported in literature (Carter et al., 2005). The NO2 photolysis rate is used to characterize UV intensity and was determined to be 0.5 min-1, similar to ambient levels (e.g. 0.53 min-1 at 0 degrees zenith angle, Carter et al., 2005)

(c) Reviewer: P3 line 31 "loss of organic vapors to Teflon surfaces" Teflon films, used for aerosol smog-chambers, are known to store various gaseous species, especially NOx, which is released from the Teflon film by UV radiation and increased temperatures. Has this been observed or taken into account? Please add a related statement to the manuscript.

Response: We have added a discussion on wall emissions and conducted chamber modeling to estimate the background contribution to secondary OH chemistry. Overall, chlorine-isoprene chemistry dominates gas-phase chemistry and secondary HOx production.

Manuscript changes in Section 2.1: "Between experiments, "blank experiments" were conducted in which seed particles, ozone, and chlorine gas (Cl2 Airgas, 106 ppm in N2) were injected into the chamber at high concentrations and UV lights were turned on to remove any residual organics that are released from the Teflon® chamber under UV. Background effects were estimated using the SAPRC chamber modeling software (http://www.engr.ucr.edu/∼carter/SAPRC/) in combination with the Carbon Bond 6 (CB6r2) chemical mechanism which was modified to include basic gas phase inorganic chlorine chemistry in addition to Cl2 and ClNO2 photolysis (Sarwar, Simon, Bhave, & Yarwood, 2012; Yarwood et al., 2010). Wall effects are represented within the model by a constant emission of nitrous acid (HONO) from the chamber walls on the order of 0.1 ppb min-1, which was determined separately in chamber characterization experiments (Carter et al., 2005)."

Manuscript changes in Section 3.4: "The SAPRC chamber model results indicate that more than 99% of the isoprene reacts with Cl; secondary OH chemistry is therefore only a very minor pathway in these experiments. Model results also show that HO2 production is dominated by isoprene-chlorine chemistry, whereas wall effects dominate HO2 production (>60 %) after all isoprene has been consumed. It is worth noting that the model does not explicitly represent Cl-initiated oxidation of reaction products, which can produce additional HOx radicals. Therefore, we expect the actual secondary OH chemistry to be more important than the current model estimation."

References

Carter, W. P. L., Cocker, D. R., Fitz, D. R., Malkina, I. L., Bumiller, K., Sauer, C. G., Pisano, J. T., Bufalino, C. and Song, C.: A new environmental chamber for evaluation of gas-phase chemical mechanisms and secondary aerosol formation, Atmos. Environ., 39, 7768–7788, 2005

Sarwar, G., Simon, H., Bhave, P., & Yarwood, G. (2012). Examining the impact of heterogeneous nitryl chloride production on air quality across the United States. Atmospheric Chemistry and Physics, 12(14), 6455–6473. doi:10.5194/acp-12-6455-2012

Yarwood, G., Jung, J., Whitten, G. Z., Heo, G., Mellberg, J., & Estes, M. (2010). Updates to the Carbon Bond Mechanism for Version 6 (CB6). In Presented at the 9th Annual CMAS Conference, Chapel Hill, NC, October 11-13 (Vol. 6, pp. 1–4).

---

## Author Comment (AC3) · 17 Aug 2017

We thank the referee for the suggestions and recommendations. Below are our responses to all comments.

1.) Reviewer: As the authors know well, acidity plays a MAJOR role for IEPOX uptake yielding isoprene SOA under low-NOx conditions. This was conclusively demonstrated with authentic IEPOX for the first time by Lin et al. (2012, ES&T); however, Wang et al. (2005, RCM), Surratt et a. (2006, JPCA), Paulot et al. (2009, Science), and Surratt et al. (2010, PNAS) were some of the first studies to propose for the existence of IEPOX even though an authentic standard did not exist at that time to study its reactive

[Figure]

uptake. Since then, kinetic studies have demonstrated that acidity plays a key role in IEPOX producing substantial amounts of SOA (Gaston et al., 2014, ES&T; Riedel et al., 2015, ES&T Letters; Riedel et al., ACP, 2016). If ammonium sulfate aerosol is wet, due to a high enough RH, then ammonium sulfate can take up IEPOX to yield SOA if the reaction time scales are long enough (Nguyen et al., 2014, ACP).

With this reminder above, I wonder why the authors did not consider also conducting experiments at elevated RH and increased acidities with the ammonium sulfate seed aerosol? I can imagine if the chemistry applies to remote marine locations, the aerosol may be more wet and/or acidic (especially if there are sufficient DMS emissions). Jon Abbatt's group also showed recently in Wong et al. (2015, ES&T) that deliquesced ammonium sulfate particles can yield a lot of SOA through a non-IEPOX route. So this could be something important to consider.

Response: We have conducted four additional experiments, two with initial chlorine injection (Exp. A6 and A7) and two with continuous chlorine injection (Exp. C6 and C7). We observed significant increases in SOA concentrations when acidified seeding aerosol was used as well as increases in higher m/z mass fragments including m/z 82, which is generally associated with IEPOX-aerosol. We also observed lower gas-phase ion signals in the CIMS in acidified seed experiments compared to neutral seed experiments. Some of the changes made to the manuscript include,

Manuscript changes in Figure 1: Added time-series from Exp. C7, which used acidified seed particles for comparison with Exp. C2, which used neutral seed particles under otherwise similar experimental conditions. The figure shows that more SOA is formed in acidified seed experiments.

Manuscript changes in Table 1: Included the results from the four additional experiments. We also updated the wall loss-corrected SOA concentration and yields using updated relative ionization efficiency (RIE) values.

Manuscript changes in Section 3.1: "The data shown in Figure 1 and summarized

in Table 1 also suggest that aerosol acidity promotes SOA formation: the SOA concentrations observed in acidified seed Exp. C7 are more than twice as high as SOA concentrations observed in neutral seed Exp. C2."

Manuscript changes in Section 3.4: "Comparison of ACSM mass spectra (see Fig. S9) shows that the presence of acidic seed aerosol increases the contribution of organic ion fragments at m/z 82 (likely C5H6O+) to the overall SOA mass (e.g. increases from 0.0055 in Exp. C2 to 0.0064 in Exp. C7; see Fig. S9), which is associated with IEPOX-derived OA (Budisulistiorini et al., 2013). Reduction in gas-phase product concentrations were also observed in the CIMS when the aerosol is acidic (see Fig. S10). These observations are consistent with increased partitioning of gas-phase products to the aerosol when the seed aerosol is acidic, resulting in the higher SOA concentrations shown in Fig. 1 and Table 1."

Added Figure S9 to supplement: Additional figure comparing the ACSM unit-mass-resolution spectra obtained from neutral and acidified seed experiments

Added Figure S10 to supplement: Additional figure comparing the CIMS unit-mass-resolution spectra obtained from neutral and acidified seed experiments

Reviewer: By the way, the authors don't appear to say how the ammonium sulfate aerosol were injected into the chamber? What was the concentration of your atomizing solution? This should be added to the experimental section.

Response: We have added additional details on our experimental protocol

Manuscript changes in Section 2.1: "Neutral seed particles were injected using an Aerosol Generation System (Brechtel, AGS Model 9200) with a 0.01 M ammonium sulfate solution; acidic seed injection used a solution containing 0.005 M ammonium sulfate and 0.0025 M sulfuric acid."

2.) Reviewer: PMF/ME-2 analyses of your SOA composition using the ACSM data:

As the authors likely know, Lin et al. (2012, ES&T), and more specifically Budisulistiorini et al. (2013, ES&T), demonstrated that AMS and ACSM, respectively, datasets can resolve IEPOX-OA factor when PMF is applied. Why didn't the authors consider conducting PMF in their analyses to constrain how much of the SOA is from IEPOX? You could run PMF with the IEPOX-OA factor constrained using the reference MS library (so this would be ME-2). Furthermore, Krechmer et al. (2015, ES&T) did this for the non-IEPOX SOA pathway. He used his reference mass spectrum for the non-IEPOX SOA to constrain its importance to field aerosol collected during the 2013 SOAS campaign! Riva et al. (2016, ES&T) also showed that authentic ISOPOOH makes SOA without needing IEPOX due to the low-volatility nature of the multifunctional hydroperoxides produced!

Since you don't use offline chemical analyses to measure molecular-level SOA components, I think it is worth while conducting PMF/ME-2 analyses to see if that can help constrain the different pathways yielding the total SOA mass. I hope the authors might agree with this suggestion.

Response: We have attempted PMF analysis (without constraining factors) on these data but were unable to extract a factor related to IEPOX-OA. This is likely because the contribution of IEPOX-OA to total SOA mass from chlorine-initiated oxidation of isoprene, if present, is small. As mentioned in the revised manuscript, the observed f82 (the fraction of the total organic signal due to ions at m/z 82) is ∼0.006. Assuming that the IEPOX-OA factor would have an f82 value of 0.0132 based on the work of Budisulistiorini et al. (2013), the contribution of the IEPOX-OA to total OA in the present study would be at most 0.5%. It is considered infeasible to extract factors with such low contributions to total OA mass; for example, Ulbrich et al. (2009) suggest that only factors which contribute at least 5% to total OA mass can be extracted reliably using PMF. We have added this information to the revised manuscript.

Manuscript changes in Section 3.4: "Lastly, considering the observed f82 value (∼0.006) and that typically associated with IEPOX-OA factors (0.013∼0.022) (Budisulistiorini et al., 2013; Krechmer et al., 2015), IEPOX-OA is estimated to contribute

less than 0.5% to the total OA formed in these experiments. This suggests that the contribution of secondary OH chemistry to SOA formation initiated by chlorine radicals is minor."

3.) Reviewer: Please go through carefully and make sure certain references are not missing throughout the text. I mention a few of these in my minor comments below.

Response: Additional references have been included.

4.) Reviewer: I know it isn't a focus in this manuscript, but it would be very powerful if molecular tracers could be identified for Cl-initiated radicals yielding SOA. The authors mention using the ACSM to try to constrain the organochlorine budget, but seemed to have trouble with this due to interference issues. This is why I suggested conducting PMF/ME-2 analyses above in # 2. However, does the CIMS data (especially the io-dide reagent ion chemistry) suggest the presence of low-volatility hydroperoxides that contain chlorine in them? From OH radical studies by Krechmer et al. (2015, ES&T), Riva et al. (2016, ES&T), and Liu et al. (2016, ES&T), they all measured low-volatility multifunctional hydroperoxides that made sufficient amounts of SOA (that don't require aerosol acidity like IEPOX).

Response: Based on our CIMS measurements, multigenerational reaction pathways that could lead to hydroperoxide formation from continued oxidation of early C5 oxi-dation products are plausible. We show time series for some of these multifunctional chloroalkyl hydroperoxides in the updated Figure 6 and added a summary of a simpli-fied suggested formation pathway in Figure 7 (an excerpt of which is attached to this response). References to previous work on hydroperoxide formation/oxidation under low NOx are added as well.

Manuscript changes in Figure 6a and 6b: Updated with chloroalkyl hydroperoxides observed in both CIMS modes that match mechanistic expectations

Manuscript changes in Figure 7: Replaced with a summary of simplified (only one

isomer is shown) reaction pathways that could give rise to the observed hydroperoxide species.

Manuscript changes in Section 3.4: "Multifunctional, low volatility hydroperoxides produced from non-IEPOX OH-isoprene reaction pathways under low NOx condition have been found to contribute to SOA formation (Krechmer et al., 2015; Liu et al., 2016; Riva et al., 2016), and the same can be expected of the chloroalkyl hydroperoxide species identified in this work including C5H7ClO3 and C5H8Cl2O3."

(5). Reviewer: When reviewing Table 1, I realized it wasn't well explained in the text why the different injection methods were used. What did these methods explicitly tell you?

Response: The different injection methods were used to separate SOA formation from the effects of vapor wall loss. We have added some clarifying text.

Manuscript changes in Section 2.1: "Initial Cl2 experiments were performed to achieve rapid oxidation of isoprene and to separate initial SOA formation from effects of vapor wall loss. Because chlorine radicals are not expected to regenerate, continuous Cl2 injection experiments were performed to provide a steady Cl radical source to control the extent of gas-phase reactions."

Manuscript changes in Section 3.1: "After extended photooxidation (T>100 mins), SOA concentrations achieved via initial chlorine injection and continuous chlorine injection differ by less than 8 $\mu$g m-3 (< 20% of total SOA mass at the time for Exp. C2), which could be attributed to vapor wall loss of early generation low-volatility products."

Reviewer: For modelers, this Table might be very difficult for them to judge which yields should be used. Also, related to my point # 1 above, modelers seeing these yields may question if these yields are accurate to remote low-NOx regions where Cl radical chemistry might matter. Can the authors offer which yield may be the most appropriate to use?

[Figure]

Response: We have added the following qualifications and recommendations.

Manuscript changes in Section 3.3: "Under atmospheric conditions, the VOC to chlorine ratio will usually be higher than those used in these experiments. Previous studies on chlorine-initiated SOA formation from toluene (Cai et al., 2008) and limonene (Cai and Griffin, 2006) suggest that SOA yield decreases with higher VOC-to-chlorine ratio. While we do not observe a clear correlation between SOA yield and isoprene-chlorine ratios used in this work (0.49-1.22), such dependence could be present over a wider range of this ratio. For air quality models, the use of the continuous case yield, which is similar to recently reported OH-oxidation yields (Liu et al., 2016; Xu et al., 2014), is more appropriate because the isoprene-to-chlorine ratio is closer to atmospheric conditions. Furthermore, the SOA yields from continuous injection experiments account for fragmentation reactions in the atmosphere (which occur throughout the experiment). Thus, yields form continuous injection experiments should be used in air-quality models which do not explicitly account for fragmentation reactions."

Reviewer: Finally, I'm assuming these various injection methods were used to gain some insights into vapor wall losses? It remains unclear to me how exactly vapor wall losses were dealt with (if at all) in reporting the SOA yields shown in Table 1.

Response: The different injection methods were indeed used to separate the effects of vapor wall loss from SOA formation, as clarified above. The wall loss correction method used accounts for depositional particle loss, as well as organic vapor loss to the deposited particles. Essentially, the correction method assumes that organic aerosols lost to the chamber wall would still participate in equilibrium partitioning as if they were suspended. Loss of organic vapor to the clean Teflon$^{®}$ surface is not accounted for here. This was described in more concise terms in the manuscript,

From Section 2.3: "Assuming internal mixing of particles and that organic vapor can condense onto suspended and wall-deposited particles alike, we corrected for particle wall loss and the loss of organic vapors onto wall-deposited particles using the organicto-sulfate ratio (Hildebrandt et al., 2009)"

Minor comments

1.) Reviewer: Abstract, Page 1, Line 12: Remove "%" after "8." You don't need this.

Response: We have removed "%"

2.) Reviewer: Methods, Page 2, Line 32: Insert a space between "exceeding100"

Response: We have fixed this typographical error.

3.) Reviewer: Section 3.4, Page 7, Line 27: You write "3-methyl-3-butene-2-one (CMBO) [(C5H7OCl)H+]." This appears to be named incorrectly. Please name according to IUPAC.

Response: We have corrected the naming to 1-chloro-3-methyl-3-butene-2-one

4.) Reviewer: Section 3.4, Page 7, Line 31: I would reference Kroll et al. (2006, ES&T) and Surratt et al. (2006, JPCA) as one of the initial references to demonstrates MACR oxidation is a source of SOA.

Response: We have modified the references as suggested.

5.) Reviewer: Section 3.4, Page 8, Line 26: The authors should reference Lin et al. (2012, ES&T)

Response: We have added the suggested reference.

6.) Reviewer: Section 3.4, Page 8, Line 27: The authors should also reference Gaston et al. (2014, ES&T) and Riedel et al. (2015, ES&T Letters)

Response: We have added the suggested references.

References

Budisulistiorini, S.H., Canagaratna, M.R., Croteau, P.L., Marth, W.J., Baumann, K., Edgerton, E.S., Shaw, S.L., Knipping, E.M., Worsnop, D.R., Jayne, J.T., Gold, A., Surratt, J.D., 2013. Real-time continuous characterization of secondary organic aerosol derived from isoprene epoxydiols in downtown Atlanta, Georgia, using the aerodyne aerosol chemical speciation monitor. Environ. Sci. Technol. 47, 5686–5694. doi:10.1021/es400023n

Cai, X., Griffin, R.J., 2006. Secondary aerosol formation from the oxidation of biogenic hydrocarbons by chlorine atoms. J. Geophys. Res. Atmos. 111, 7348–7359. doi:10.1029/2005JD006857

Cai, X., Ziemba, L.D., Griffin, R.J., 2008. Secondary aerosol formation from the oxidation of toluene by chlorine atoms. Atmos. Environ. 42, 7348–7359. doi:10.1016/j.atmosenv.2008.07.014

Hildebrandt, L., Donahue, N.M., Pandis, S.N., 2009. High formation of secondary organic aerosol from the photo-oxidation of toluene. Atmos. Chem. Phys. 9, 2973–2986. doi:10.5194/acp-9-2973-2009

Krechmer, J.E., Coggon, M.M., Massoli, P., Nguyen, T.B., Crounse, J.D., Hu, W., Day, D.A., Tyndall, G.S., Henze, D.K., Rivera-Rios, J.C., Nowak, J.B., Kimmel, J.R., Mauldin, R.L., Stark, H., Jayne, J.T., Sipila, M., Junninen, H., St. Clair, J.M., Zhang, X., Feiner, P.A., Zhang, L., Miller, D.O., Brune, W.H., Keutsch, F.N., Wennberg, P.O., Seinfeld, J.H., Worsnop, D.R., Jimenez, J.L., Canagaratna, M.R., 2015. Formation of Low Volatility Organic Compounds and Secondary Organic Aerosol from Isoprene Hydroxyhydroperoxide Low-NO Oxidation. Environ. Sci. Technol. 49, 10330–10339. doi:10.1021/acs.est.5b02031

Liu, J., D'Ambro, E.L., Lee, B.H., Lopez-Hilfiker, F.D., Zaveri, R.A., Rivera-Rios, J.C., Keutsch, F.N., Iyer, S., Kurten, T., Zhang, Z., Gold, A., Surratt, J.D., Shilling, J.E., Thornton, J.A., 2016. Efficient Isoprene Secondary Organic Aerosol Formation from a Non-IEPOX Pathway. Environ. Sci. Technol. 50, 9872–9880. doi:10.1021/acs.est.6b01872

Riva, M., Budisulistiorini, S.H., Chen, Y., Zhang, Z., D'Ambro, E.L., Zhang, X., Gold, A., Turpin, B.J., Thornton, J.A., Canagaratna, M.R., Surratt, J.D., 2016. Chemical Characterization of Secondary Organic Aerosol from Oxidation of Isoprene Hydroxy-hydroperoxides. Environ. Sci. Technol. 50, 9889–9899. doi:10.1021/acs.est.6b02511

Ulbrich, I.M., Canagaratna, M.R., Zhang, Q., Worsnop, D.R., Jimenez, J.L., 2009. Interpretation of Organic Components from Positive Matrix Factorization of Aerosol Mass Spectrometric Data. Atmos. Chem. Phys. 9, 2891. doi:10.5194/acp-9-2891-2009

Xu, L., Kollman, M.S., Song, C., Shilling, J.E., Ng, N.L., 2014. Effects of NOx on the volatility of secondary organic aerosol from isoprene photooxidation. Environ. Sci. Technol. 48, 2253–2262. doi:10.1021/es404842g

[Figure]

**Fig. 1.** Excerpt from Figure 7 showing potential hydroperoxide formation pathways

---

## Author Comment (AC4) · 17 Aug 2017

We thank the referee for the suggestions and recommendations. Below are our responses to all comments.

1. Reviewer: Introduction: In motivating their laboratory study, the authors may wish to point out marine emissions of isoprene and the consideration of isoprene SOA from this source (e.g. Gantt et al 2010, Atmos. Environ.). The authors currently do not consider the potential for Cl oxidation of isoprene far from coasts (where NOx may also be low), which is a motivating factor for this work.

[Figure]

Response: We have added discussions of SOA formation from marine emissions of isoprene, sources of halogen emissions, and Cl oxidation of isoprene in continental regions, to the introduction of the revised manuscript. While the model results by Gantt et al. (2010) suggest that contribution of marine isoprene emissions to coastal SOA (and O3) loadings are small, we note that the isoprene SOA yield used in the CMAQ model (3 % for low NOx) is lower than values (5∼15 %) reported in more recent literature (Krechmer et al., 2015; Liu et al., 2016; Riva et al., 2016). The inclusion of additional emissions of halogen species and findings from this work could reveal much greater contributions of coastal isoprene chemistry to total OA loading. The findings are also applicable to continental isoprene chemistry under low NOx conditions.

2. Reviewer: Throughout the manuscript, chlorine incorporated into organic molecules appears to be referred to as "chloride", which chemically refers to Cl-, rather than chloro-organics, or organic chlorine. This reference to chloride is confusing as it makes the reader question whether the authors are indeed suggesting that inorganic chloride is present in the particle phase. This needs to be clarified throughout.

Response: In the context of chlorine-initiated oxidation of isoprene, the term (organic) "chloride" refers to the "-Cl" functional group in chloro-organics found in the condensed phase. To avoid confusion, we have changed all references to particulate organic chlorine to "organochlorides". This is more in line with other categorizations such as "organonitrate" or "organosulfate".

3. Reviewer: Sec 2.2, Instrumentation, should be clarified in the main text in terms of the description of the CIMS, for which additional information is needed. Perhaps material from the supplemental information should be moved here, in addition to revisions for clarity. It is stated that proton transfer, charge exchange, and clustering are all used for chemical ionization, which is confusing since typically one pathway is chosen through specific conditions within the ion molecule region of the instrument. As worded, it sounds like these reaction pathways of analyte ion formation are all occurring simultaneously.

Response: We have added some operational details about the CIMS and modified/clarified the original statement. Clustering is the chemical ionization mechanism within the ion-molecule reaction (IMR) region. Declustering can occur between the IMR and Time-of-Flight region and the resulting ion products may appear as if they were generated via proton transfer or charge transfer.

Manuscript changes in Section 2.2: "Reagent ions are generated by passing humidified UHP N2 over a methyl iodide permeation tube and then through a 210Po radioactive cartridge (NRD, 2013) at 2 LPM into the ion-molecule reaction (IMR) region operating at 200 mbar pressure. Analyte, "M" can undergo chemical ionization within the IMR:

(H2O)n(H3O+) + M → (H2O)n(MH+) + H2O (1)

(H2O)n(I-) + M → (H2O)n(MI-) (2)

depending on which ion mode is selected. The number of clusters, "n" ranges from 0 and 2 for (H2O)n(H3O+), with (H2O)(H3O+) being the most dominant reagent ion in the positive ion mode. Hydronium-water cluster CIMS was used to detect isoprene and select moderately oxidized species. For (H2O)n(I-) ionization, "n" ranges from 0 to 1, with I- being the most dominant reagent ion. Water-iodide cluster CIMS was used to detect select highly oxidized and acidic species (Aljawhary et al., 2013; Lee et al., 2014)."

Reviewer: It is also odd to me that the instrument doesn't seem to have been tuned for conditions of primarily H3O+, rather than (H2O)nH3O+. What fraction of the signal was associated with H3O+, and how many n were observed? This would impact the resulting analyte ionization. When were (H2O)nH3O+ vs (H2O)nI- reagent ions used? Did this switch back and forth during experiments, or was one ion chemistry used per experiment? Were CIMS experiments conducted during all experiments, or only during C3 and C5?

Response: The "hydronium CIMS" used here is different from PTR-MS in that significant reagent ion clustering is still observed even after tuning. A drift tube replacement of the ion-molecule reaction region is required to decluster reagent ions and to generate primarily H3O+ (Yuan et al., 2016). In this work the dominant reagent ions were (H2O)(H3O+) (>70% of total reagent ion signals) and I- (>80%) in positive and negative ion modes, respectively. CIMS data were collected during most experiments listed in Table 1. We experimented with mode switching between negative and positive reagent ion modes, but it was ultimately impractical due to the time required for voltage discharge and signal stabilization following each mode switch.

4. Reviewer: Page 5, First Paragraph: Was a decrease in reagent ion signal observed compared to below an experiment? This might suggest a non-linear response and concern that reagent ion reactions could be limited even if still in excess. Without calibration of the signals, this would make trends more difficult to assess if in a non-linear regime. The phrasing on lines 5-6 about this is not clear.

Response: Reduction in reagent ion signal was observed in all photooxidation experiments. As described in section S3, the instrument response is still approximately linear when the reagent ion decrease is small, in which case normalization against the dominant reagent ion should correct for signal reduction. The demonstration of normalization error for large reagent ion depletion shown in section S3 was not performed for any particular set of experimental data, but rather for a simple hypothetical case.

Reviewer: Also, why wasn't at least isoprene calibrated for since each experiment started with a known mole ratio? It seems like that would be beneficial to this work and could probably even be done retroactively with knowledge of the experimental parameters. Was "significant depletion of reagent ion" (Page S3, Line 24) observed during any experiment?

Response: We have performed a few isoprene calibration experiments in positive mode, which were not reported here. A linear response was observed. Because complete isoprene depletion was achieved in every experiment and calibration experiments

showed a linear response, it was not deemed necessary to routinely calibrate for isoprene as isoprene concentrations can be calculated based on relative signal changes over the course of each experiment.

Larger reagent ion depletion was observed in negative ionization mode during some initial chlorine injection experiments, including experiments A5 and A8, where the initial Cl2 concentration is high. (H2O)n(I-) ionization is very sensitive towards Cl2. CIMS data from initial chlorine injection experiments were not used for qualitative interpretations.

5. Reviewer: In the results and discussion, it would often be helpful, when possible, to give values in parentheses, rather than vague descriptors so that you don't require the reader to review and correctly interpret the graphs.

Response: We have added additional quantitative interpretations throughout the manuscript. For example, we have estimated the contribution of IEPOX-OA to the total OA mass: (f82 is the fractional contribution to organic aerosol mass from ions at mass-to-charge ratio 82; it is associated with IEPOX-OA factors, Budisulistiorini et al., 2013).

Manuscript changes in Section 3.4: "Lastly, considering the observed f82 value ($\sim$0.006) and that typically associated with IEPOX-OA factors (0.013$\sim$0.022) (Budisulistiorini et al., 2013; Krechmer et al., 2015), IEPOX-OA is estimated to contribute less than 0.5% to the total OA formed in these experiments. This suggests that the contribution of secondary OH chemistry to SOA formation initiated by chlorine radicals is minor."

6. Reviewer: Section 3.2 should either be moved after section 3.4 or moved to the supplemental information. This section does not contribute much to our understanding of Cl-SOA or precursors, as it primarily focuses on an issue with the ACSM method, which while important, doesn't seem to be the main focus of this work.

Response: During additional gas-phase CIMS data analysis we identified several chloroalkyl hydroperoxides that may be important for SOA growth. We have proposed reaction mechanisms and updated figures. Given this new context, we feel that the inclusion of section 3.2 is in line with the focus of the paper.

Reviewer: Rather than only identifying a potential issue, could a chlorinated organic standard be purchased and aerosolized for characterization so that the authors could provide a solution to the problem as well? Similarly, nearly a full paragraph in the conclusions is dedicated to this subject, which detracts from the exciting science studied. Also, use of m/z 36 here is not intuitive when referring to chloride.

Response: Issues with inorganic chloride or organochloride detection using ACSM and similar instrumentation (i.e. Aerosol Mass Spectrometer, "AMS") are well known. Our analysis presents evidence for particulate organochloride formation and suggests the cause of the quantification issue. It is beyond the scope of this paper to solve the issue. We plan to continue contributing to a resolution of this issue in future work on organochloride quantification including the use of standard compounds. While the representation may not be intuitive, the HCl+ ion appears to be a good proxy for particulate organochlorides.

7. Reviewer: Page 7, Lines 11-12: It is not clear, as written, if you then used a 2D model here.

Response: We did not run the 2-D model here. We added the following clarifying text,

Manuscript changes in Section 3.3: "Two-dimensional modeling would be more appropriate in these cases (Chuang and Donahue, 2016; Donahue et al., 2012; Murphy et al., 2012) but was not performed on this dataset."

8. Reviewer: Please review rules for significant figures for numbers and fix throughout. Please note that when reporting error only one significant figure should be used, with the same number of decimal places used for the average and the error.

[Figure]

Response: We have updated the significant figures and numbers

9. Reviewer: Can you compare ACSM mass spectra at different points during an experiment to examine possible evidence of oxidative fragmentation or vapor wall loss (as discussed on Page 7, Lines 18-19)? Could ACSM mass spectra be compared between experiments to examine the potential for differences in SOA composition?

Response: Most noticeable changes over the course of the experiment are observed at m/z 44 and m/z 43. The fractional contributions to the total organic mass by ions at m/z 44 ("f44") and 43 ("f43"), which are used to construct the triangle plot shown in Figure 3, can be used to estimate the oxidation state of carbon as described in section 2.3 and S2. The estimated SOA oxidation state and its trend are shown in Fig 2. Evidence of vapor wall loss is shown in Figure S1, where a decrease in SOA concentration was observed in the dark (in the absence of photooxidation effects). Additionally, we have conducted four new experiments using acidified seed particles. Comparisons of ACSM mass spectra between neutral vs. acidified seed aerosol experiments have been added to the revised manuscript/supplement (Fig. S9).

10. Reviewer: A conclusion of the study is that "The effects of SOA aging must be described explicitly and separately from initial SOA formation." (Page 9, lines 16-17) Yet, few details are given in the results and discussion for what this explicit description is. Above I suggested possible ways to provide greater mechanistic information on the Cl-isoprene oxidation and subsequent SOA formation.

Response: The purpose of the referenced statement was to reinforce the notion that a 1-D VBS framework cannot adequately describe SOA decay (due to fragmentation reactions) following SOA formation. The 2D-VBS model or more mechanistic frameworks are needed to describe that behavior. In this work, initial Cl2 injection experiments were performed to separate initial SOA formation from effects of vapor wall loss and fragmentation. We have added clarifications in the revised manuscript. As mentioned above in response to comment 6, we have proposed some reaction pathways that are

consistent with gas-phase observations, SOA formation, and organochloride detection to provide more mechanistic information on isoprene-chlorine reactions as suggested.

Manuscript changes in Section 3.4: "Multifunctional, low volatility hydroperoxides produced from non-IEPOX OH-isoprene reaction pathways under low NOx conditions have been found to contribute to SOA formation (Krechmer et al., 2015; Liu et al., 2016; Riva et al., 2016), and the same can be expected of the chloroalkyl hydroperoxide species identified in this work including C5H7ClO3 and C5H8Cl2O3."

Manuscript changes in Section 4. "Initial chlorine injection experiment results show that initial SOA formation is followed by SOA decay, driven by oxidative fragmentation and vapor wall loss, which cannot be adequately described by a 1-D VBS model."

11. Reviewer: Another conclusion of the study is "Similarities between chlorine-isoprene and OH-isoprene oxidation products suggest that air quality models may be able to lump the treatment of SOA produced from chlorine- and OH-initiated oxidation of isoprene." Yet this is difficult to discern as very little discussion was dedicated to this important topic. There also appears to be no quantitative information that would indicate similar yields associated with various reaction pathways. More in-depth interpretation and discussion of the data is required to support this statement.

Response: Figure 5 shows that the chlorine-isoprene SOA yields are similar to OH-isoprene SOA yields under low NOx conditions. Figure 3 shows that the chlorine-isoprene SOA have similar f43 and f44 as OH-isoprene SOA, indicating that they are similarly oxidized. Many air quality models parameterize SOA formation using either constant yields or yields that depend on organic aerosol loading, for example, using 1-dimensional volatility basis set (VBS) parameterizations (Donahue et al., 2006) or 2-product models (Odum et al., 1996). A few models have also incorporated the OA oxidation state using 2D (or 1.5D) VBS representations (Donahue et al., 2011; Koo et al., 2014). In any case, most SOA modeling efforts focus on yields and oxidation state of OA, not the detailed chemical mechanisms. Considering that OA yields and

oxidation state are similar for chlorine and OH-initiated oxidation of isoprene, the same 1D or 2D VBS parameterizations could be used to represent isoprene SOA formation initiated by these two oxidants. While there are some similarities in the types of product species observed (e.g. chloroalkyl hydroperoxides), detailed chemical models such as the master chemical mechanism would need to treat OH and Cl-initiated oxidation of isoprene separately considering the difference in reaction pathways. Much more work is needed to fully understand SOA formation from Cl-initiated oxidation of isoprene, and the continued atmospheric processing of OA, which is beyond the scope of this paper. This has been clarified in the revised manuscript, in which we now also offer recommendations for modelers on what yields to use for chlorine-initiated oxidation of isoprene.

Manuscript changes in section 4: "For air quality models which do not explicitly account for SOA aging and fragmentation reactions, the averaged SOA yield from continuous chlorine injection experiments should be used (8 %), which accounts for oxidative fragmentation effects in the atmosphere."

Manuscript changes in section 4: "Proposed reaction mechanisms and gas-phase measurements by CIMS show that Cl-initiated oxidation of isoprene could produce chloroalkyl hydroperoxide species, analogous to the formation of low-volatility hydroperoxides observed for OH-isoprene oxidation under low NOx conditions, which may explain the high yields observed."

Responses to minor comments

-Reviewer: Page 2, Line 11: It is unclear why Riedel et al 2012 is cited here, since that is a coastal marine study.

Response: This reference has been removed.

- Reviewer: Page 4, Line 12: It is confusing to have several equations on the same line. It would be preferable to have one equation per line and number as such.

Response: The equations are now individually numbered.

- Reviewer: Page 4, Lines 17-18: The words "low" and "high" are vague, and it would be useful to include at least approximate values in parentheses as well, for example, to aid in interpretation of these descriptors.

Response: The ratio of total organic aerosol loading to volatility bin saturation concentration is more important than the respective absolute values. When the ratio is 20, roughly 95% of the organic mass would partition to the particle phase (for that volatility bin). Further increase in the ratio would see diminishing returns in terms of SOA yield, and a maximum yield is achieved.

-Reviewer: Page 6, lines 4-5: This sentence states "high reactivity of chlorine radicals toward isoprene and its reaction products" and therefore seems to contradict the earlier sentence on page 5, lines 27-28.

Response: According to Orlando et al. (2002), the bimolecular rate constant of the isoprene-chlorine reaction is $4.3 \pm 0.6$ x 10-10 cm3 molecule-1 s-1, whereas the rate constants of the MVK-chlorine and MACR-chlorine reactions are $2.2 \pm 0.3$ x 10-10 and $2.4 \pm 0.3$ x 10-10 cm3 molecule-1 s-1, respectively. The earlier statement about high isoprene concentrations does not necessarily contradict high reactivity of chlorine with volatile organic compounds.

- Reviewer: Page 6, Line 10: The phrase "quantification proved to be difficult" is vague.

Response: This phrase has been removed.

- Reviewer: Page 7, Lines 13-14: I would suggest deleting this sentence, as the previous paragraph already explained this and having this information here as well could confuse the reader.

Response: This sentence has been removed.

- Reviewer: Page 7, Line 15: For clarity, I suggest adding the following to the end of

the sentence". . .literature values of OH oxidation under low and high NOx scenarios."

Response: We have edited the sentence as suggested.

- Reviewer: Page 7, Line 16: Why was the highest observed SOA yield reported, rather than an average, for example?

Response: The values reported in Table 1 are maximum wall-loss corrected SOA yield values from each experiment. These maximum values are then averaged to calculate the average yield value for all continuous chlorine injection experiments (average yield for initial chlorine experiments is calculated in a similar fashion separately from continuous experiments). The maximum SOA concentration coincides with the time of complete isoprene depletion for continuous chlorine injection experiments, which is used as a reference condition for reporting yields. This has been clarified in the revised manuscript.

- Reviewer: Page 7, Line 16: By "continuous cases", do you mean continuous Cl2 injection during an experiment? Make sure this is clear.

Response: Yes. Continuous cases refer to experiments during which Cl2 is continuously injected. This has been clarified in the revised manuscript.

- Reviewer: Hypochlorous acid is generally written as HOCl. I'm not used to ClOH, as written throughout.

Response: We have changed the representation to HOCl throughout.

- Reviewer: Page 9, Lines 1-2: This sentence is commenting on the method, more so than the science and could be moved to the methods section or supplemental.

Response: We have removed this sentence.

- Reviewer: Page 9, Line 16: It isn't clear why this sentence is needed to be highlighted in the conclusions.

Response: We have removed this sentence.

- Reviewer: Page 9, Lines 18-24: It would be useful to merge this short paragraph with the first paragraph of the conclusions section.

Response: We have merged the first and second paragraphs of section 4.

- Reviewer: Figure 3 caption: This figure does not explicitly show oxidation state as stated in the caption, which is misleading.

Response: The caption has been updated to clarify that the plotted parameters are often used as a proxy of the oxidation state.

- Reviewer: Figure 5 caption: For clarification, I suggest adding the phrase "corresponding to low and high NOx OH oxidation" at the end of the first sentence.

Response: We have clarified the caption as suggested.

- Reviewer: Figure 6 caption: It is not clear what is meant by "interfering ions" here.

Response: This figure and caption have been updated in the revised manuscript. In the discussion version of the manuscript, the figure caption should have read "(C5H10O3)H+ and interfering ions, (C5H8O2)H+ and (C5H11O3Cl)H+." referring to the possibility of (C5H10O3)H+ being a adduct product (C5H8O2)(H3O)+, or an ion fragment of another chlorinated ion.

- Reviewer: S1: Please provide references for this section of the SI.

Response: To our knowledge, the derivations shown are new. Starting equations are from works by Donahue et al. (2006)

- Reviewer: Page S3, Line 13: Why would instrument sensitivity change over time?

Response: Change in instrument sensitivity over the course of an experiment is caused by changes in the concentrations of reagent ions available for chemical ionization. Over time, instrument conditions may change, such as deterioration of the micro-channel

plate (MCP) or ToF pressures.

- Reviewer: Page S3, Line 16: k does not appear to be defined.

Response: "k" is the collision rate constant. This has been clarified in the revised manuscript.

References

Aljawhary, D., Lee, a. K.Y., Abbatt, J.P.D., 2013. High-resolution chemical ionization mass spectrometry (ToF-CIMS): Application to study SOA composition and processing. Atmos. Meas. Tech. 6, 3211–3224. doi:10.5194/amt-6-3211-2013

Chuang, W.K., Donahue, N.M., 2016. A two-dimensional volatility basis set; Part 3: Prognostic modeling and NOx dependence. Atmos. Chem. Phys. 16, 123–134. doi:10.5194/acp-16-123-2016

Donahue, N.M., Epstein, S.A., Pandis, S.N., Robinson, A.L., 2011. A two-dimensional volatility basis set: 1. organic-aerosol mixing thermodynamics. Atmos. Chem. Phys. 11, 3303–3318. doi:10.5194/acp-11-3303-2011

Donahue, N.M., Robinson, A.L., Stanier, C.O., Pandis, S.N., 2006. Coupled partitioning, dilution, and chemical aging of semivolatile organics. Environ. Sci. Technol. 40, 2635–2643. doi:10.1021/es052297c

Gantt, B., Meskhidze, N., Zhang, Y., Xu, J., 2010. The effect of marine isoprene emissions on secondary organic aerosol and ozone formation in the coastal United States. Atmos. Environ. 44, 115–121. doi:10.1016/j.atmosenv.2009.08.027

Koo, B., Knipping, E., Yarwood, G., 2014. 1.5-Dimensional volatility basis set approach for modeling organic aerosol in CAMx and CMAQ. Atmos. Environ. 95, 158–164. doi:10.1016/j.atmosenv.2014.06.031

Krechmer, J.E., Coggon, M.M., Massoli, P., Nguyen, T.B., Crounse, J.D., Hu, W., Day, D.A., Tyndall, G.S., Henze, D.K., Rivera-Rios, J.C., Nowak, J.B., Kimmel, J.R.,

Mauldin, R.L., Stark, H., Jayne, J.T., Sipila, M., Junninen, H., St. Clair, J.M., Zhang, X., Feiner, P.A., Zhang, L., Miller, D.O., Brune, W.H., Keutsch, F.N., Wennberg, P.O., Seinfeld, J.H., Worsnop, D.R., Jimenez, J.L., Canagaratna, M.R., 2015. Formation of Low Volatility Organic Compounds and Secondary Organic Aerosol from Isoprene Hydroxyhydroperoxide Low-NO Oxidation. Environ. Sci. Technol. 49, 10330–10339. doi:10.1021/acs.est.5b02031

Lee, B.H., Lopez-Hilfiker, F.D., Mohr, C., Kurtén, T., Worsnop, D.R., Thornton, J.A., 2014. An iodide-adduct high-resolution time-of-flight chemical-ionization mass spectrometer: Application to atmospheric inorganic and organic compounds. Environ. Sci. Technol. 48, 6309–6317. doi:10.1021/es500362a

Murphy, B.N., Donahue, N.M., Fountoukis, C., Dall'Osto, M., O'Dowd, C., Kiendler-Scharr, A., Pandis, S.N., 2012. Functionalization and fragmentation during ambient organic aerosol aging: Application of the 2-D volatility basis set to field studies. Atmos. Chem. Phys. 12, 10797–10816. doi:10.5194/acp-12-10797-2012

Odum, J.R., Hoffmann, T., Bowman, F., Collins, D., Flagan Richard, C., Seinfeld John, H., 1996. Gas particle partitioning and secondary organic aerosol yields. Environ. Sci. Technol. 30, 2580–2585. doi:10.1021/es950943+

Orlando, J.J., Tyndall, G.S., Apel, E.C., Riemer, D.D., Paulson, S.E., 2003. Rate coefficients and mechanisms of the reaction of Cl-atoms with a series of unsaturated hydrocarbons under atmospheric conditions. Int. J. Chem. Kinet. 35, 334–353. doi:10.1002/kin.10135

Riva, M., Budisulistiorini, S.H., Chen, Y., Zhang, Z., D'Ambro, E.L., Zhang, X., Gold, A., Turpin, B.J., Thornton, J.A., Canagaratna, M.R., Surratt, J.D., 2016. Chemical Characterization of Secondary Organic Aerosol from Oxidation of Isoprene Hydroxy-hydroperoxides. Environ. Sci. Technol. 50, 9889–9899. doi:10.1021/acs.est.6b02511

Xu, L., Kollman, M.S., Song, C., Shilling, J.E., Ng, N.L., 2014. Effects of NOx on the

volatility of secondary organic aerosol from isoprene photooxidation. Environ. Sci. Technol. 48, 2253–2262. doi:10.1021/es404842g

Yuan, B., Koss, A., Warneke, C., Gilman, J.B., Lerner, B.M., Stark, H., De Gouw, J.A., 2016. A high-resolution time-of-flight chemical ionization mass spectrometer utilizing hydronium ions (H3O+ ToF-CIMS) for measurements of volatile organic compounds in the atmosphere. Atmos. Meas. Tech. 9, 2735–2752. doi:10.5194/amt-9-2735-2016

---

## Author Comment (AC5) · 14 Sep 2017

An error was made in the response to question (2) by anonymous referee #4. The question, original author comment, and corrected author comment are shown below.

Question 2.) Reviewer: PMF/ME-2 analyses of your SOA composition using the ACSM data:

As the authors likely know, Lin et al. (2012, ES&T), and more specifically Budisulistior-ini et al. (2013, ES&T), demonstrated that AMS and ACSM, respectively, datasets can resolve IEPOX-OA factor when PMF is applied. Why didn't the authors consider con-
ducting PMF in their analyses to constrain how much of the SOA is from IEPOX? You could run PMF with the IEPOX-OA factor constrained using the reference MS library (so this would be ME-2). Furthermore, Krechmer et al. (2015, ES&T) did this for the non-IEPOX SOA pathway. He used his reference mass spectrum for the non-IEPOX SOA to constrain its importance to field aerosol collected during the 2013 SOAS campaign! Riva et al. (2016, ES&T) also showed that authentic ISOPOOH makes SOA without needing IEPOX due to the low-volatility nature of the multifunctional hydroperoxides produced!

Since you don't use offline chemical analyses to measure molecular-level SOA components, I think it is worth while conducting PMF/ME-2 analyses to see if that can help constrain the different pathways yielding the total SOA mass. I hope the authors might agree with this suggestion.

-Error in the original author response: "We have attempted PMF analysis (without constraining factors) on these data but were unable to extract a factor related to IEPOX-OA. This is likely because the contribution of IEPOX-OA to total SOA mass from chlorine-initiated oxidation of isoprene, if present, is small. As mentioned in the revised manuscript, the observed f82 (the fraction of the total organic signal due to ions at m/z 82) is ∼0.006. Assuming that the IEPOX-OA factor would have an f82 value of 0.0132 based on the work of Budisulistiorini et al. (2013), the contribution of the IEPOX-OA to total OA in the present study would be at most 0.5%. It is considered infeasible to extract factors with such low contributions to total OA mass; for example, Ulbrich et al. (2009) suggest that only factors which contribute at least 5% to total OA mass can be extracted reliably using PMF."

-Corrected author comment: By this line of logic, the maximum IEPOX-OA contribution would actually be 50 % (not 0.5 %), which is unrealistic. We note that the observed f82 values (0.0055 to 0.0064) are lower than the average value observed for ambient OA that has been influenced by isoprene emissions (0.0065 ± 0.0022) (Hu et al., 2015), which would suggest that IEPOX-OA did not contribute significantly to SOA formation

(Wong et al., 2015). In other chlorine-initiated SOA formation from biogenic volatile organic compounds experiments, the observed f82 values were in some cases as high as 0.005, which is close to the observed f82 values for isoprene-chlorine SOAs. Updated Figures 6 and 7 show that $C_5H_8O_2$, which could produce $C_5H_6O^+$ fragment within the ACSM, was observed as a gas-phase product. It is plausible that other biogenic SOA species would produce ion fragments at m/z 82 ($C_5H_6O^+$ or other ions with the same integer m/z). A more detailed discussion is included in the revised manuscript.

[revised manuscript text omitted]

Wong, J.P.S., Lee, A.K.Y., Abbatt, J.P.D., 2015. Impacts of Sulfate Seed Acidity and Water Content on Isoprene Secondary Organic Aerosol Formation. Environ. Sci. Technol. 49, 13215–13221. doi:10.1021/acs.est.5b02686

---

## Author Comment (AC6) · 14 Sep 2017

An error was made in an example given as part of the response to question (5) by anonymous referee #2. The question, original author comment, and corrected author comment are shown below.

-Question 5, Reviewer: In the results and discussion, it would often be helpful, when possible, to give values in parentheses, rather than vague descriptors so that you don't require the reader to review and correctly interpret the graphs.

-Error in the original author response: "Lastly, considering the observed f82 value

(∼0.006) and that typically associated with IEPOX-OA factors (0.013∼0.022) (Budisulistiorini et al., 2013; Krechmer et al., 2015), IEPOX-OA is estimated to contribute less than 0.5% to the total OA formed in these experiments. This suggests that the contribution of secondary OH chemistry to SOA formation initiated by chlorine radicals is minor."

-Corrected author comment: By this line of logic, the maximum IEPOX-OA contribution to chlorine-isoprene SOA would be 50 % (not 0.5 %), which is unrealistic. We note that the observed f82 values (0.0055 to 0.0064) are lower than the average value observed for ambient OA that has been influenced by isoprene emissions (0.0065 ± 0.0022) (Hu et al., 2015), which would suggest that IEPOX-OA did not contribute significantly to SOA formation (Wong et al., 2015). In other chlorine-initiated SOA formation from biogenic volatile organic compounds experiments, the observed f82 values were in some cases as high as 0.005, which is close to the observed f82 values for isoprene-chlorine SOAs. Updated Figures 6 and 7 show that $C_5H_8O_2$, which could produce $C_5H_6O^+$ fragment within the ACSM, was observed as a gas-phase product. It is plausible that other biogenic SOA species would produce ion fragments at m/z 82 ($C_5H_6O^+$ or other ions with the same integer m/z). A more detailed discussion on IEPOX and secondary OH chemistry is included in the revised manuscript.

[revised manuscript text omitted]

Wong, J.P.S., Lee, A.K.Y., Abbatt, J.P.D., 2015. Impacts of Sulfate Seed Acidity and Water Content on Isoprene Secondary Organic Aerosol Formation. Environ. Sci. Technol. 49, 13215–13221. doi:10.1021/acs.est.5b02686

---

## Author Response (AR1)

Author's Response

We thank the referees for the suggestions and recommendations. Responses to all comments start on the next page. A marked-up version of the revised manuscript is attached at the end.

Major comments and responses.

(a) Reviewer: *p.5, line 1: The limitations of estimating O:C, H:C, and the oxidation state of carbon from f44 based on empirical correlations should be briefly discussed. For example, equation S-1 in section S2 may underestimate O:C values substantially in environments dominated by NOx-free isoprene chemistry (Canagaratna et al., 2015). Also, the presence of heteroatoms may introduce deviations from equation S-3 in section S2 when estimating the oxidation state of carbon (Kroll et al., 2011).*

Response: We have added a discussion of the limitations of the empirical correlation and ACSM data to the revised manuscript. We have also updated the equation labels. Equation S2 is now Eq.S10, and Eq.S3 is now Eq.S11.

Section 2.3, Pg.6, Line 14-17: "The empirical correlations were derived using a comprehensive collection of Aerosol Mass Spectrometer datasets but may underestimate O:C values for SOA formed under low $NO_x$ conditions from isoprene or toluene (Canagaratna et al., 2015). Variability in $f_{43}$ and $f_{44}$ among different ACSMs have also been reported. (Crenn et al., 2015)."

Section S2, Pg.2, Line 8-10: "Deviation from Eq. (S11) could occur due to the presence of peroxide groups or heteroatom groups (Kroll et al., 2011), such as chloroalkyl hydroperoxide compounds identified in CIMS measurements."

(b) Reviewer: *p.5, line 13: The separation of experiments A1-A5 and experiments H1/H2 seems to be somewhat arbitrary. In my opinion, experiment H1 should be experiment A6, and the much higher maximum yield of this experiment should be part of the discussion of section 3.3. Experiment H2 is a technical experiment to "...explore the ability of the ACSM to detect organic chloride" (supplement, section S4). I was confused to find information about H2 in Table 1, and I recommend to remove it from the table and just explain the character of this experiment in the supplement.*

Response: We agree with the referee's recommendation. We have re-designated Exp. H1 as Exp. A8 and have conducted two additional initial chlorine injection experiments which are now designated as A6 and A7. Experiment H2 is now designated as Exp. S1. Experimental details for Exp. S1 are now described in-text within supplement section S4.

(c) Reviewer: *p.5, line 21: Chlorinated organic compounds have also been identified in ambient aerosol samples from Western Australia by ion cyclotron mass spectrometry (Kamilli et al., 2016), with a higher abundance of chlorinated organic compounds in daytime samples when photochemistry is active.*

Response: References to this work have been added to section 1 and section 3.2.

d) Reviewer: *p.7, line 19: The VOC:Cl2 ratios may be expected to be much higher under atmospheric conditions than in the presented experiments. Do the authors have some insight, or could they speculate about how the yields may change for larger isoprene:Cl2 ratios? Also, when presenting the highest observed SOA yields, why do the authors exclude experiment A1 for the average yield of the initial injection experiments?*

Response: We have added discussion on a potential correlation between the VOC:Cl$_2$ ratio and SOA yields, as well as details that may be of interest to modelers. The yield calculated from experiment A1 was in fact used in calculating the average initial chlorine experiment SOA yields in the discussion paper. The text "A2-A5" was a typographical error and has now been corrected to "A1-A5." Based on new calibration results we have also updated the relative ionization efficiency (RIE) values used in ACSM data analysis, which resulted in lower calculated yield values.

Section 3.3, Pg.9, Line 23-32: "Under atmospheric conditions, the isoprene-chlorine ratio will usually be higher than ratios used in these experiments. Previous studies on chlorine-initiated SOA formation from toluene (Cai et al., 2008) and limonene (Cai and Griffin, 2006) suggest that SOA yields decrease with higher VOC-to-Cl ratio. While we do not observe a clear correlation between SOA yield and isoprene-to-chlorine ratios used here (0.5-1.2), such dependence could be present over a wider ratio range. For air quality models which do not explicitly account for fragmentation reactions, the use of the average continuous case yield, which is similar to recently reported OH-oxidation yields (Liu et al., 2016; Xu et al., 2014), is more appropriate because the isoprene-to-chlorine ratio is closer to atmospheric conditions and because the SOA yields from continuous injection experiments account for effects of OA aging in the atmosphere (which occur throughout the experiments). The presence of acidic aerosols and inclusion of particulate chlorine content would increase expected yields."

e) Reviewer: *p.8, line 8: When discussing secondary OH chemistry, the authors mention potentially unidentified HOx production pathways other than HO2 production during formation of CMBO. It would be extremely interesting to have at least a semi-quantitative estimate of the contributions of chlorine-initiated secondary OH chemistry vs. OH chemistry from other sources, potentially also due to chamber wall effects.*

Response: We have carried out some chamber box modeling and expanded the discussion of HO$_x$ chemistry. Model results show that chlorine chemistry accounts for the majority of HO$_2$ production when sufficient isoprene is present. Overall, chlorine radicals consume over 99% of the isoprene. Oxidation of isoprene by OH radical has very minor contribution in these experiments.

Section 2.1, Pg. 3 Line 29–Pg.4 Line 2: "Background effects have been quantified using chamber characterization experiments (Carter et al., 2005) and the SAPRC chamber modeling software (http://www.engr.ucr.edu/~carter/SAPRC/) in combination with the Carbon Bond 6 (CB6r2) chemical mechanism, which was modified to include basic gas phase inorganic chlorine

chemistry in addition to $Cl_2$ and $ClNO_2$ photolysis (Sarwar et al., 2012; Yarwood et al., 2010). $NO_x$-offgasing is represented within the model by a constant emission of nitrous acid (HONO) from the chamber walls on the order of 0.1 ppb $min^{-1}$, which was determined separately in chamber characterization experiments (Carter et al., 2005)."

5   Section 3.4, Pg. 10, Line 20-21: "The formation of CMBO and MBO produces $HO_2$ radicals (Orlando et al., 2003; Ragains and Finlayson-Pitts, 1997), which serve as a source of secondary OH radicals. Other $RO_2 + RO_2$ reaction pathways also produce $HO_x$ radicals."

Section 3.4, Pg.11, Line 18-24: "For chlorine-initiated oxidation of isoprene, the SAPRC chamber model results indicate that over 99% of the isoprene reacts with Cl; OH oxidation of isoprene is therefore only a very minor pathway in these experiments. Model results also show that $HO_2$ production is dominated by isoprene-chlorine chemistry when sufficient isoprene is present, whereas wall effects dominate $HO_2$ production ($> 60\%$) after all isoprene has been consumed. The model does not explicitly represent Cl-initiated oxidation of reaction products, which can produce additional $HO_x$ radicals, and therefore likely underestimates the importance of secondary OH chemistry."

Technical Comments and Responses

*-Reviewer: in manuscript: p.2, line 16 and p.7, line.27: When referring to isomers of CMBO, these should be isomers of chloromethylbutenone, e.g. 1-chloro-3-methyl-3-butene-2-one, not "isomers of 3-methyl-3-butene-2-one".*

Response: We have corrected the naming. See Pg.3, line 4.

*-Reviewer: p.3, line 23: Change "relatively ionization efficiencies" to "relative ionization efficiencies".*

Response: We have corrected the typographical error. See Pg.5, Line 1.

*-Reviewer: p.4, line 10: The reference should read "Odum et al., 1996".*

25   Response: We have corrected the reference. See Pg.5, Line 21.

*-Reviewer: p.9, line 4: Change "produced form" to "produced from". p.9, line 6: Change "chlorine-initiation oxidation" to "chlorine-initiated oxidation".*

Response: This sentenced has been removed in the revised manuscript.

*-Reviewer: p.18, Table 1: I don't understand the value of the VOC:Cl2 ratio in experiment H2.*

Response: Wrong precursor concentrations were reported for Exp. H2 (now Exp. S1). We have corrected the errors in the SI.

*-Reviewer: p.3, line 7: Change "number of water cluster" to "number of water clusters". p.4, line 7: Remove "greater than" before "44 % overestimation could be expected".*

Response: This sentence has been removed in the revised manuscript.

Major comments and responses

*1. Reviewer: Introduction: In motivating their laboratory study, the authors may wish to point out marine emissions of isoprene and the consideration of isoprene SOA from this source (e.g. Gantt et al 2010, Atmos. Environ.). The authors currently do not*
5 *consider the potential for Cl oxidation of isoprene far from coasts (where NOx may also be low), which is a motivating factor for this work.*

Response: We have added discussions of SOA formation from marine emissions of isoprene, halogen sources and observations (e.g. reactive chlorine species within the marine boundary layer) to the introduction of the revised manuscript to motivate the
10 work. While the model results by Gantt et al. (2010) suggest that contribution of marine isoprene emissions to coastal SOA (and $O_3$) loadings are small, we note that the isoprene SOA yield used in the CMAQ model (3 % for low $NO_x$) is lower than values (5~15 %) reported in more recent literature (Krechmer et al., 2015; Liu et al., 2016; Riva et al., 2016). The inclusion of additional emissions of halogen species and findings from this work could reveal much greater contributions of coastal isoprene chemistry to total OA loading. We agree with the referee that the findings are also applicable to continental isoprene chemistry
15 (and, in addition, photochemistry over salt lakes) under low NOx conditions.

Section 1, Pg.1 Line 28 - Pg.2 Line 6: "Isoprene SOA formation initiated by ozone ($O_3$) , nitrate, and hydroxyl (OH) radicals has been studied extensively and is estimated to account for 6–30 Tg yr$^{-1}$ of the global aerosol budget (Brégonzio-Rozier et al., 2015; Claeys, 2004; Guenther et al., 2006; Kroll et al., 2006; Lin et al., 2012; Surratt et al., 2010, 2006; Zhao et al., 2015),
20 but the importance of isoprene SOA formation within the marine boundary layer (MBL) remains highly disputed in literature (Arnold et al., 2009; Bikkina et al., 2014; Fu et al., 2011, 2013; Gantt et al., 2015, 2010; Hu et al., 2013; Luo and Yu, 2010; O'Dowd and de Leeuw, 2007). Although production of reactive chlorine species such as $Cl_2/HOCl$ has been observed within the MBL (Lawler et al., 2011), little is known about SOA from chlorine-initiated oxidation of isoprene."

25 Section 1, Pg.2, Line 7-23: Overview of tropospheric sources and sinks for reactive halogenic species. Please refer to the responses to question (a) by referee #3 on Pg.16 in this document.

*2. Reviewer: Throughout the manuscript, chlorine incorporated into organic molecules appears to be referred to as "chloride", which chemically refers to Cl-, rather than chloro-organics, or organic chlorine. This reference to chloride is*
30 *confusing as it makes the reader question whether the authors are indeed suggesting that inorganic chloride is present in the particle phase. This needs to be clarified throughout.*

Response: In the context of chlorine-initiated oxidation of isoprene, the term (organic) "chloride" refers to the "-Cl" functional group in chloro-organics found in the condensed phase. We have clarified particle-phase chlorine quantification using ACSM

in the revised text. To avoid confusion, we have changed all references to particulate organic chlorine to "organochlorides". This is more in line with other categorizations such as "organonitrate" or "organosulfate". We have also added a discussion on the potential contribution to measured particulate chlorine contents from inorganic chlorides.

5    Section 3.2, Pg 8, Line 4-6: "Aerosol analytes undergo electron impact ionization in the ACSM, and chlorine-containing ion fragments are mostly expected at $m/z$ 35 and 37 as $Cl^+$ and at $m/z$ 36 and 38 as $HCl^+$. Larger organochloride ion fragments may exist but cannot be separated in the unit-mass resolution spectra."

Section 3.2, Pg 8, Line 6-12: "A previous study on chorine-initiated oxidation of toluene, which proceeds primarily through a
10    hydrogen-abstraction pathway, reported particulate chlorine formation (4 % of the total aerosol mass), which was attributed to HCl uptake (Cai et al., 2008). Formation of organochloride aerosol has been observed previously for chlorine-initiated oxidation of alpha-pinene (Ofner et al., 2013), which proceeds primarily via a chlorine-addition reaction pathway. Thus, isoprene-chlorine reactions are expected to result in particulate organochloride formation. The uptake of HCl produced from Cl H-abstraction or intramolecular HCl elimination (Ragains and Finlayson-Pitts, 1997) could also contribute slightly to
15    observed particulate chlorine."

*3. Reviewer: Sec 2.2, Instrumentation, should be clarified in the main text in terms of the description of the CIMS, for which additional information is needed. Perhaps material from the supplemental information should be moved here, in addition to revisions for clarity. It is stated that proton transfer, charge exchange, and clustering are all used for chemical ionization,*
20    *which is confusing since typically one pathway is chosen through specific conditions within the ion molecule region of the instrument. As worded, it sounds like these reaction pathways of analyte ion formation are all occurring simultaneously.*

Response: We have added some operational details about the CIMS and modified/clarified the original statement. Clustering is the chemical ionization mechanism within the ion-molecule reaction (IMR) region. Declustering can occur between the IMR
25    and Time-of-Flight region and the resulting ion products may appear as if they were generated via proton transfer or charge transfer.

Section 2.2, Pg.4, Line 17-27: "Reagent ions are generated by passing humidified UHP $N_2$ over a methyl iodide permeation tube and then through a $^{210}$Po radioactive cartridge (NRD, 2013) at 2 LPM into the ion-molecule reaction (IMR) chamber
30    operating at 200 mbar pressure. Analyte, "M" can undergo chemical ionization within the IMR with hydronium-water ($[H_2O]_n[H_3O]^+$) or iodide-water ($[H_2O]_nI^-$) ion clusters,

$$(H_2O)_n(H_3O)^+ + M \rightarrow (H_2O)_n(MH)^+ + H_2O \qquad (1)$$
$$(H_2O)_nI^- + M \rightarrow (H_2O)_n(MI)^- \qquad (2)$$

where the number of clusters, "n" ranges from 0 and 2 for $[H_2O]_n[H_3O]^+$, with $[H_2O][H_3O]^+$ being the most dominant reagent ion in the positive ion mode. Hydronium-water cluster CIMS was used to detect isoprene and select moderately oxidized species (Aljawhary et al., 2013). For $[H_2O]_nI^-$ ionization, "n" ranges from 0 to 1, with $I^-$ being the most dominant reagent ion. Water-iodide cluster CIMS was used to detect select highly oxidized and acidic species (Aljawhary et al., 2013; Lee et al.,

5     2014)."

*Reviewer: It is also odd to me that the instrument doesn't seem to have been tuned for conditions of primarily $H_3O^+$, rather than $(H_2O)_nH_3O^+$. What fraction of the signal was associated with $H_3O^+$, and how many n were observed? This would impact the resulting analyte ionization. When were $(H_2O)_nH_3O^+$ vs $(H_2O)_nI^-$ reagent ions used? Did this switch back and forth during*
10   *experiments, or was one ion chemistry used per experiment? Were CIMS experiments conducted during all experiments, or only during C3 and C5?*

Response: The "hydronium CIMS" used here is different from PTR-MS in that significant reagent ion clustering is still observed even after tuning. A drift tube replacement of the ion-molecule reaction region is required to decluster reagent ions and to generate primarily $H_3O^+$ reagent ions (Yuan et al., 2016). In this work, the dominant reagent ions were $(H_2O)(H_3O^+)$
15   (>70% of total reagent ion signals) and $I^-$ (>80%) in positive and negative ion modes, respectively.  CIMS data were collected during most experiments listed in Table 1. We experimented with mode switching between negative and positive reagent ion modes, but it was ultimately impractical due to the time required for voltage discharge and signal stabilization following each mode switch.

*4. Reviewer: Page 5, First Paragraph: Was a decrease in reagent ion signal observed compared to below an experiment? This might suggest a non-linear response and concern that reagent ion reactions could be limited even if still in excess. Without calibration of the signals, this would make trends more difficult to assess if in a non-linear regime. The phrasing on lines 5-6 about this is not clear.*

Response: Reduction in reagent ion signal was observed in all photooxidation experiments. As described in section S3, the instrument response is still approximately linear when the reagent ion signal decrease is small, in which case normalization against the dominant reagent ion should correct for signal reduction. The demonstration of normalization error for large reagent ion depletion shown in section S3 was not performed for any particular set of experimental data, but rather for a simple
30   hypothetical case.

*Reviewer: Also, why wasn't at least isoprene calibrated for since each experiment started with a known mole ratio? It seems like that would be beneficial to this work and could probably even be done retroactively with knowledge of the experimental parameters. Was "significant depletion of reagent ion" (Page S3, Line 24) observed during any experiment?*

Response: We have performed a few isoprene calibration experiments in positive mode, which were not reported here. A linear response was observed. Because complete isoprene depletion was achieved in every experiment and calibration experiments showed a linear response, it was not deemed necessary to routinely calibrate for isoprene as isoprene concentrations can be calculated based on relative signal changes over the course of each experiment. Larger reagent ion depletion was observed in negative ionization mode during some initial chlorine injection experiments, including experiments A5 and A8, where the initial $Cl_2$ concentration is high. $(H_2O)_n(I^-)$ ionization is very sensitive towards $Cl_2$. CIMS data from initial chlorine injection experiments were not used for qualitative interpretations.

*5. Reviewer: In the results and discussion, it would often be helpful, when possible, to give values in parentheses, rather than vague descriptors so that you don't require the reader to review and correctly interpret the graphs.*

Response: We have added additional quantitative interpretations throughout the manuscript. We note that there was an error regarding the interpretation of $f_{82}$ value in the original author response posted. We have revised our analysis. The scientific conclusion about the contribution of IEPOX-OA to observed SOA formation was not affected.

Section 3.1, Pg.7, Line 3-6: "The initial chlorine experiment (A4) exhibited similar trends, but the SOA decay was faster, where 30 µg m$^{-3}$ SOA decay (40% of maximum SOA mass) occurred within about 30 minutes of photo-oxidation ($9 < T < 40$ mins), likely due to more rapid oxidation and fragmentation of reaction products."

Section 3.1, Pg.7, Line 9-11: "After extended photo-oxidation ($T > 100$ mins), SOA concentrations achieved via initial chlorine injection (Exp. A4) and continuous chlorine injection (Exp. C2) differed by less than 8 µg m$^{-3}$ ($< 20\%$ of total SOA mass at T = 100 mins for Exp. C2)."

Section 3.4, Pg12, Line 2-11: A discussion on the importance of IEPOX is provided. Please refer to responses to question (2) by referee #4 on Pg.20-21 in this document for more details.

*6. Reviewer: Section 3.2 should either be moved after section 3.4 or moved to the supplemental information. This section does not contribute much to our understanding of Cl-SOA or precursors, as it primarily focuses on an issue with the ACSM method, which while important, doesn't seem to be the main focus of this work.*

Response: During additional gas-phase CIMS data analysis we identified several chloroalkyl hydroperoxides that may be important for SOA growth. We have proposed reaction pathways and expanded our gas-phase chemistry discussion. Please see the revised Section 3.4, updated Figures 6 and 7. Given this new context, we feel that the inclusion of section 3.2 is in line with the focus of the paper.

*Reviewer: Rather than only identifying a potential issue, could a chlorinated organic standard be purchased and aerosolized for characterization so that the authors could provide a solution to the problem as well? Similarly, nearly a full paragraph in the conclusions is dedicated to this subject, which detracts from the exciting science studied. Also, use of m/z 36 here is not intuitive when referring to chloride.*

Response: Issues with inorganic chloride or organochloride detection using ACSM and similar instrumentation (i.e. Aerosol Mass Spectrometer, "AMS") are well known. Our analysis presents evidence for particulate organochloride formation and suggests the cause of the quantification issue. It is beyond the scope of this paper to solve the issue. We plan to continue contributing to a resolution of this issue in future work on organochloride quantification including the use of standard

10 compounds. While the representation may not be intuitive, the $HCl^+$ ion appears to be a good proxy for particulate organochlorides.

*7. Reviewer: Page 7, Lines 11-12: It is not clear, as written, if you then used a 2D model here.*

15 Response: We did not run the 2-D model here. We added the following clarifying text,

Section 3.3, Pg.9 Line 13-15: "Two-dimensional modeling would be more appropriate in these cases (Chuang and Donahue, 2016; Donahue et al., 2012; Murphy et al., 2012) but was not performed on this dataset."

20 *8. Reviewer: Please review rules for significant figures for numbers and fix throughout. Please note that when reporting error only one significant figure should be used, with the same number of decimal places used for the average and the error.*

Response: We have updated the significant figures and numbers

25 *9. Reviewer: Can you compare ACSM mass spectra at different points during an experiment to examine possible evidence of oxidative fragmentation or vapor wall loss (as discussed on Page 7, Lines 18-19)? Could ACSM mass spectra be compared between experiments to examine the potential for differences in SOA composition?*

Response: Most noticeable changes over the course of the experiment are observed at *m/z* 44 and *m/z* 43. The fractional

30 contributions to the total organic mass by ions at *m/z* 44 ("*f*44") and 43 ("*f*43"), which are used to construct the triangle plot shown in Figure 3, can be used to estimate the oxidation state of carbon as described in section 2.3 and S2. The estimated SOA oxidation state and its trend are shown in Fig 2. Evidence of vapor wall loss is shown in Figure S1, where a decrease in SOA concentration was observed in the dark (in the absence of photo-oxidation effects). Additionally, we have conducted four new

experiments using acidified seed particles. Comparisons of ACSM mass spectra between neutral vs. acidified seed aerosol experiments have been added to the revised manuscript in section 3.4 and S7. See also Figure S13 and S14.

Section 3.4, Pg12, Line 2-11: A discussion on the importance of IEPOX is provided based on ACSM and CIMS unit mass resolution spectra comparison. Please refer to responses to question (2) by referee #4 on Pg.20-21 in this document for more details.

*10. Reviewer: A conclusion of the study is that "The effects of SOA aging must be described explicitly and separately from initial SOA formation." (Page 9, lines 16-17) Yet, few details are given in the results and discussion for what this explicit description is. Above I suggested possible ways to provide greater mechanistic information on the Cl-isoprene oxidation and subsequent SOA formation.*

Response: The purpose of the referenced statement was to reinforce the notion that a 1-D VBS framework cannot adequately describe SOA decay (due to fragmentation reactions) following SOA formation. An example is provided in Section S5. The 2D-VBS model or more mechanistic frameworks are needed to describe that behavior. In this work, initial $Cl_2$ injection experiments were performed to separate initial SOA formation from effects of vapor wall loss and fragmentation. We have added clarifications in the revised manuscript. As mentioned above in response to comment 6, we have proposed some formation pathways (see Section S6) that are consistent with gas-phase observations, SOA formation, and organochloride detection to provide more mechanistic information on isoprene-chlorine reactions as suggested.

Section 3.4, Pg.11, Line 3-9: "For instance, the $C_5H_8ClO_2{}^{\bullet}$ radical produced via Cl-addition to isoprene could either undergo $RO_2 + RO_2$ chemistry to produce $C_5H_7ClO$ (e.g. CMBO) or undergo $RO_2 + HO_2$ chemistry to produce $C_5H_9ClO_2$, a chloroalkyl hydroperoxide. Similarly, the $C_5H_7O_2{}^{\bullet}$ radical produced via H-abstraction from isoprene could undergo $RO_2 + RO_2$ chemistry to produce $C_5H_6O$ (e.g. MBO) or undergo $RO_2 + HO_2$ chemistry to produce $C_5H_8O_2$, a hydroperoxide. Ions consistent with $C_5H_9ClO_2$ and $C_5H_8O_2$ are observed in the gas-phase, as shown in Fig 6a and 6b, where the formation of $RO_2 + HO_2$ reaction products appears delayed compared to their $RO_2 + RO_2$ pathway counterparts. This is consistent with $RO_2 + RO_2$ reactions being a source of $HO_x$ radicals."

Section 4, Pg. 12: The sentence referenced by the referee has been removed, as the point has been addressed in section 3.3 and S5.

*11. Reviewer: Another conclusion of the study is "Similarities between chlorine-isoprene and OH-isoprene oxidation products suggest that air quality models may be able to lump the treatment of SOA produced from chlorine- and OH-initiated oxidation of isoprene." Yet this is difficult to discern as very little discussion was dedicated to this important topic. There also appears*

*to be no quantitative information that would indicate similar yields associated with various reaction pathways. More in-depth interpretation and discussion of the data is required to support this statement.*

Response: Similarities between chlorine-isoprene and OH-isoprene SOAs can be seen in particle-phase measurements. Gas-
5    phase oxidation products also share some overlaps. Figure 5 shows that the chlorine-isoprene SOA yields are similar to OH-isoprene SOA yields under low NO$_x$ conditions. Figure 3 shows that the chlorine-isoprene SOA have similar $f_{43}$ and $f_{44}$ as OH-isoprene SOA, indicating that they are similarly oxidized. Many air quality models parameterize SOA formation using either constant yields or yields that depend on organic aerosol loading, for example, using 1-dimensional volatility basis set (VBS) parameterizations (Donahue et al., 2006) or 2-product models (Odum et al., 1996). A few models have also incorporated the
10   OA oxidation state using 2D (or 1.5D) VBS representations (Donahue et al., 2011; Koo et al., 2014). In any case, most SOA modeling efforts focus on yields and oxidation state of OA, not the detailed chemical mechanisms. Considering that OA yields and oxidation state are similar for chlorine and OH-initiated oxidation of isoprene, the same 1D or 2D VBS parameterizations could be used to represent isoprene SOA formation initiated by these two oxidants. While there are some similarities in the types of product species observed (e.g. chloroalkyl hydroperoxides), detailed chemical models such as the master chemical
15   mechanism would need to treat OH- and Cl-initiated oxidation of isoprene separately considering the difference in reaction pathways. Much more work is needed to fully understand SOA formation from Cl-initiated oxidation of isoprene, and the continued atmospheric processing of OA, which is beyond the scope of this paper. This has been revised in the manuscript, in which we now also offer recommendations for modelers on what yields to use for chlorine-initiated oxidation of isoprene.

20   Section 4, Pg. 12, Line 19-21: "For air quality models which do not explicitly account for SOA aging, the averaged SOA yield from continuous chlorine injection experiments (8%) should be used for SOA formation from chlorine-initiated oxidation of isoprene."

Section 4, Pg. 12, Line 30-32: "Proposed formation pathways and gas-phase measurements by the CIMS show that Cl-initiated
25   oxidation of isoprene could produce chloroalkyl hydroperoxide species, analogous to the formation of low-volatility hydroperoxides observed for OH-isoprene oxidation under low NO$_x$ conditions."

Minor comments and responses
*-Reviewer: Page 2, Line 11: It is unclear why Riedel et al 2012 is cited here, since that is a coastal marine study.*
30   Response: This reference has been removed.

*- Reviewer: Page 4, Line 12: It is confusing to have several equations on the same line. It would be preferable to have one equation per line and number as such.*
Response: The equations are now individually numbered as Eq.(5)–(8).

*- Reviewer: Page 4, Lines 17-18: The words "low" and "high" are vague, and it would be useful to include at least approximate values in parentheses as well, for example, to aid in interpretation of these descriptors.*

Response: The ratio of total organic aerosol loading to volatility bin saturation concentration is more important than the respective absolute values. When the ratio is 20, roughly 95% of the organic mass would partition to the particle phase (for

5    that volatility bin). Further increase in the ratio would see diminishing returns in terms of SOA yield, and a maximum yield is achieved.

*-Reviewer: Page 6, lines 4-5: This sentence states "high reactivity of chlorine radicals toward isoprene and its reaction products" and therefore seems to contradict the earlier sentence on page 5, lines 27-28.*

10    Response: According to Orlando et al. (2002), the bimolecular rate constant of the isoprene-chlorine reaction is determined to be $4.3 \pm 0.6 \times 10^{-10}$ cm$^3$ molecule$^{-1}$ s$^{-1}$, whereas the rate constants of the MVK-chlorine and MACR-chlorine reactions are $2.2 \pm 0.3 \times 10^{-10}$ and $2.4 \pm 0.3 \times 10^{-10}$ cm$^3$ molecule$^{-1}$ s$^{-1}$, respectively. The earlier statement about high isoprene concentrations does not necessarily contradict high reactivity of chlorine with volatile organic compounds.

15    *- Reviewer: Page 6, Line 10: The phrase "quantification proved to be difficult" is vague.*

Response: This phrase has been removed.

*- Reviewer: Page 7, Lines 13-14: I would suggest deleting this sentence, as the previous paragraph already explained this and having this information here as well could confuse the reader.*

20    Response: This sentence has been removed.

*- Reviewer: Page 7, Line 15: For clarity, I suggest adding the following to the end of the sentence"…literature values of OH oxidation under low and high NOx scenarios."*

Response: We have edited the sentence as suggested. See section 3.3, Pg. 9, Line 19.

*- Reviewer: Page 7, Line 16: Why was the highest observed SOA yield reported, rather than an average, for example?*

Response: The values reported in Table 1 are maximum wall-loss corrected SOA yield values from each experiment. These maximum values are then averaged to calculate the average yield value for all continuous chlorine injection experiments (average yield for initial chlorine experiments is calculated in a similar fashion separately from continuous experiments). The

30    maximum SOA concentration coincides with the time of complete isoprene depletion for continuous chlorine injection experiments, which is used as a reference condition for reporting yields. This has been clarified in the revised manuscript.

Section 3.3, Pg. 9, Line 16-17: "Complete isoprene depletion, which coincides with the maximum SOA concentration (see Fig. 1), is used as the reference condition for yield reporting in Table 1."

*- Reviewer: Page 7, Line 16: By "continuous cases", do you mean continuous Cl2 injection during an experiment? Make sure this is clear.*

Response: Yes. Continuous cases refer to experiments during which Cl2 is continuously injected. This has been clarified in the revised manuscript.

Section 3.4, Pg. 9. Line 22-23: "Observed SOA yields averaged $20 \pm 3$ % for initial $Cl_2$ injection cases (A1–A5) and $8 \pm 1$ % for continuous $Cl_2$ injection cases (C1–C4)."

*- Reviewer: Hypochlorous acid is generally written as HOCl. I'm not used to ClOH, as written throughout.*

10 Response: We have changed the representation to HOCl throughout.

*- Reviewer: Page 9, Lines 1-2: This sentence is commenting on the method, more so than the science and could be moved to the methods section or supplemental.*

Response: This sentence has been reworked as part of the discussion on the potential IEPOX contribution to SOA.

*- Reviewer: Page 9, Line 16: It isn't clear why this sentence is needed to be highlighted in the conclusions.*

Response: We have removed this sentence.

*- Reviewer: Page 9, Lines 18-24: It would be useful to merge this short paragraph with the first paragraph of the conclusions*

20 *section.*

Response: We have merged the first and second paragraphs of section 4.

*- Reviewer: Figure 3 caption: This figure does not explicitly show oxidation state as stated in the caption, which is misleading.*

Response: The caption has been updated to "extent of oxidation".

*- Reviewer: Figure 5 caption: For clarification, I suggest adding the phrase "corresponding to low and high NOx OH oxidation" at the end of the first sentence.*

Response: We have clarified the caption as suggested.

30 *- Reviewer: Figure 6 caption: It is not clear what is meant by "interfering ions" here.*

Response: This figure and caption have been updated in the revised manuscript. In the discussion version of the manuscript, the figure caption should have read "$(C_5H_{10}O_3)H^+$ and interfering ions, $(C_5H_8O_2)H^+$ and $(C_5H_{11}O_3Cl)H^+$." referring to the possibility of $(C_5H_{10}O_3)H^+$ being a adduct product $(C_5H_8O_2)(H_3O)^+$, or an ion fragment of another chlorinated ion.

*- Reviewer: S1: Please provide references for this section of the SI.*

Response: To our knowledge, the derivations shown are new. Starting equations are from works by Donahue et al. (2006)

*- Reviewer: Page S3, Line 13: Why would instrument sensitivity change over time?*

5    Response: Change in instrument sensitivity over the course of an experiment is caused by changes in the concentrations of reagent ions available for chemical ionization. Over time, instrument conditions may change, such as deterioration of the micro-channel plate (MCP) or ToF pressures.

*- Reviewer: Page S3, Line 16: k does not appear to be defined.*

10    Response: "k" is the collision rate constant. This has been clarified in the revised S.I.

Major comments and responses

(a) Reviewer: *Introduction: The authors should add a Paragraph to the introduction about natural and anthropogenic halogen sources and sinks in the atmosphere to introduce this topic to the readers; e.g. by: Simpson et al., Tropospheric Halogen*

5  *Chemistry: Sources, Cycling, and Impacts, Chem. Reviews, 2015. Roland von Glasow, Wider role for airborne chlorine, nature, 464, 2010. Finlayson-Pitts, Halogens in the Troposphere, Anal. Chem., 82, 770-776, 2010. Buxmann et al., Consumption of reactive halogen species from sea-salt aerosol by secondary organic aerosol: slowing down the bromine explosion, Environ. Chem., 12, 476-488, 2015.*

10  Response: We have added a discussion on natural and anthropogenic halogen sources and sinks to the introduction of the revised manuscript as suggested by the reviewer.

Section 1, Pg. 2, Line 7-23: "Chlorine chemistry is known to have important effects on ozone layer depletion (Crutzen, 1974; Molina and Rowland, 1974). Recent laboratory studies and field measurements also suggest an important role of halogen (X)

15  chemistry on tropospheric composition (Faxon and Allen, 2013; Finlayson-Pitts, 2010; Saiz-Lopez and von Glasow, 2012; Simpson et al., 2015). Reactive halogen species in the form of $X_2$, XO, HOX, $XNO_2$, OXO are present in polar regions (Buys et al., 2013; Liao et al., 2014; Pöhler et al., 2010), the MBL (Lawler et al., 2011; Read et al., 2008), coastal and inland regions (Mielke et al., 2013; Riedel et al., 2013, 2012). Outside of MBL and polar regions, natural emissions of reactive halogen species have been observed in volcano plumes (Bobrowski et al., 2007) and over salt lakes (Kamilli et al., 2016; Stutz, 2002).

20  Anthropogenic sources include industrial emissions (Chang and Allen, 2006; Riedel et al., 2013; Tanaka et al., 2003), oil and gas production (Edwards et al., 2014), water treatment (Chang et al., 2001), biomass burning (Lobert et al., 1999), engine exhaust (Osthoff et al., 2008; Parrish et al., 2009), and $NO_x$-mediated heterogenous reactions, notably the production of $ClNO_2$ via reactive uptake of $N_2O_5$ onto particles containing $Cl^-$ (Thornton et al., 2010). Recent studies have found that models under-predict the abundance of reactive halogen species, suggesting incomplete understanding of their sources (Faxon et al., 2015;

25  Faxon and Allen, 2013; Simpson et al., 2015; Thornton et al., 2010). Photolysis of reactive halogen species produces halogen radicals that can react with $O_3$, hydrocarbons, SOA, and other radicals in the atmosphere. Reactions with hydrocarbons and organic aerosol serve as chlorine and bromine radical sinks (Buxmann et al., 2015; Ofner et al., 2012; Platt and Hönninger, 2003), especially in high $NO_x$ environments where halogen recycling via $HO_x$ and XO reaction pathways is suppressed (Edwards et al., 2013; Riedel et al., 2014; Simpson et al., 2015)."

(b) Reviewer: *p2 line 30: Please add the characteristics of the UVA light source: actinic flux, quantified UV/VIS spectrum.*

Response: We have included additional information on the UV light source.

Section 2.1, Pg.3, Line 22-24: "The UV spectrum is similar to other blacklight sources reported in literature (Carter et al., 2005). The $NO_2$ photolysis rate is used to characterize UV intensity and was determined to be 0.5 min$^{-1}$, similar to ambient levels (e.g. 0.53 min$^{-1}$ at 0 degrees zenith angle, Carter et al., 2005)."

5   (c) Reviewer: *P3 line 31 "loss of organic vapors to Teflon surfaces" Teflon films, used for aerosol smog-chambers, are known to store various gaseous species, especially NOx, which is released from the Teflon film by UV radiation and increased temperatures. Has this been observed or taken into account? Please add a related statement to the manuscript.*

Response: We have added a discussion on wall emissions and conducted chamber modeling to estimate the background
10   contribution to secondary $HO_x$ chemistry. Overall, chlorine-isoprene chemistry dominates gas-phase chemistry and secondary $HO_x$ production.

Section 2.1, Pg.3 Line 26 – Pg.4 Line 2: "Between experiments, 'blank experiments' were conducted in which seed particles, ozone, and chlorine gas ($Cl_2$ Airgas, 106 ppm in $N_2$) were injected into the chamber at high concentrations and UV lights were
15   turned on to remove any residual organics which could be released from the Teflon® chamber surface. Background effects have been quantified using chamber characterization experiments (Carter et al., 2005) and the SAPRC chamber modeling software (http://www.engr.ucr.edu/~carter/SAPRC/) in combination with the Carbon Bond 6 (CB6r2) chemical mechanism, which was modified to include basic gas phase inorganic chlorine chemistry in addition to $Cl_2$ and $ClNO_2$ photolysis (Sarwar et al., 2012; Yarwood et al., 2010). $NO_x$-offgasing is represented within the model by a constant emission of nitrous acid
20   (HONO) from the chamber walls on the order of 0.1 ppb min$^{-1}$, which was determined separately in chamber characterization experiments (Carter et al., 2005)."

Section 3.4, Pg.11, Line 18-24: "For chlorine-initiated oxidation of isoprene, the SAPRC chamber model results indicate that over 99% of the isoprene reacts with Cl; OH oxidation of isoprene is therefore only a very minor pathway in these experiments.
25   Model results also show that $HO_2$ production is dominated by isoprene-chlorine chemistry when sufficient isoprene is present, whereas wall effects dominate $HO_2$ production (> 60 %) after all isoprene has been consumed. The model does not explicitly represent Cl-initiated oxidation of reaction products, which can produce additional $HO_x$ radicals, and therefore likely underestimates the importance of secondary OH chemistry."

Author Response to Anonymous Referee #4

Major comments and responses

*1.) Reviewer: As the authors know well, acidity plays a MAJOR role for IEPOX uptake yielding isoprene SOA under low-NOx conditions. This was conclusively demonstrated with authentic IEPOX for the first time by Lin et al. (2012, ES&T); however,*

5 *Wang et al. (2005, RCM), Surratt et a. (2006, JPCA), Paulot et al. (2009, Science), and Surratt et al. (2010, PNAS) were some of the first studies to propose for the existence of IEPOX even though an authentic standard did not exist at that time to study its reactive uptake. Since then, kinetic studies have demonstrated that acidity plays a key role in IEPOX producing substantial amounts of SOA (Gaston et al., 2014, ES&T; Riedel et al., 2015, ES&T Letters; Riedel et al., ACP, 2016). If ammonium sulfate aerosol is wet, due to a high enough RH, then ammonium sulfate can take up IEPOX to yield SOA if the reaction time scales*

10 *are long enough (Nguyen et al., 2014, ACP).*

*With this reminder above, I wonder why the authors did not consider also conducting experiments at elevated RH and increased acidities with the ammonium sulfate seed aerosol? I can imagine if the chemistry applies to remote marine locations, the aerosol may be more wet and/or acidic (especially if there are sufficient DMS emissions). Jon Abbatt's group also showed*

15 *recently in Wong et al. (2015, ES&T) that deliquesced ammonium sulfate particles can yield a lot of SOA through a non-IEPOX route. So this could be something important to consider.*

Response: We have conducted four additional experiments, two with initial chlorine injection (Exp. A6 and A7) and two with continuous chlorine injection (Exp. C6 and C7). We observed significant increases in SOA concentrations when acidified

20 seeding aerosol was used as well as increases in some higher $m/z$ mass fragments including $m/z$ 82, which is generally associated with IEPOX-aerosol. We also observed lower gas-phase ion signals in the CIMS in acidified seed experiments compared to neutral seed experiments. Some of the changes made to the manuscript include,

Figure 1: Added time-series from Exp. C7, which used acidified seed particles for comparison with Exp. C2, which used

25 neutral seed particles under otherwise similar experimental conditions. The figure shows that more SOA is formed in acidified seed experiments.

Table 1: Included the results from the four additional experiments. We also updated the wall loss-corrected SOA concentration and yields using updated relative ionization efficiency (RIE) values.

Section 3.1, Pg.7, Line 11-13: "The data shown in Fig. 1 and summarized in Table 1 also suggest that aerosol acidity promotes SOA formation: the SOA concentration observed in acidified seed Exp. C7 were more than twice as high as SOA concentrations observed in neutral seed Exp. C2."

Section 3.4: Added a discussion of the effect of acidified seed aerosol. Please see responses to question (2) below.

Figure S13: Additional figure showing the difference ACSM unit-mass-resolution spectra obtained from neutral (Exp. C2) and acidified seed (Exp. C7) experiments.

Figure S14: Additional figure comparing the CIMS unit-mass-resolution spectra obtained from neutral (C3 and C5) and acidified seed (C6 and C7) experiments.

*Reviewer: By the way, the authors don't appear to say how the ammonium sulfate aerosol were injected into the chamber? What was the concentration of your atomizing solution? This should be added to the experimental section.*

Response: We have added additional details on our experimental protocol as follows:

Section 2.1, Pg. 4, Line 6-8: "Neutral seed particles were injected using an Aerosol Generation System (Brechtel, AGS Model 9200) with a 0.01 M ammonium sulfate solution; acidic seed particles were generated using a solution containing 0.005 M ammonium sulfate and 0.0025 M sulfuric acid."

2.) Reviewer: *PMF/ME-2 analyses of your SOA composition using the ACSM data:*
*As the authors likely know, Lin et al. (2012, ES&T), and more specifically Budisulistiorini et al. (2013, ES&T), demonstrated that AMS and ACSM, respectively, datasets can resolve IEPOX-OA factor when PMF is applied. Why didn't the authors consider conducting PMF in their analyses to constrain how much of the SOA is from IEPOX? You could run PMF with the IEPOX-OA factor constrained using the reference MS library (so this would be ME-2). Furthermore, Krechmer et al. (2015, ES&T) did this for the non-IEPOX SOA pathway. He used his reference mass spectrum for the non-IEPOX SOA to constrain its importance to field aerosol collected during the 2013 SOAS campaign! Riva et al. (2016, ES&T) also showed that authentic ISOPOOH makes SOA without needing IEPOX due to the low-volatility nature of the multifunctional hydroperoxides produced!*

*Since you don't use offline chemical analyses to measure molecular-level SOA components, I think it is worth while conducting PMF/ME-2 analyses to see if that can help constrain the different pathways yielding the total SOA mass. I hope the authors might agree with this suggestion.*

Response: We have attempted PMF analysis (without constraining factors) on these data but were unable to extract a factor related to IEPOX-OA. This is likely because the contribution of IEPOX-OA to total SOA mass from chlorine-initiated oxidation of isoprene, if present, is small. We note that an error was made in the original author response posted regarding $f_{82}$

and the estimated contribution by IEPOX-SOA. We have revised our analysis. The conclusion remains that IEPOX-OA likely did not contribute significantly to SOA formation.

Error in the original author response: "As mentioned in the revised manuscript, the observed $f_{82}$ (the fraction of the total organic signal due to ions at m/z 82) is ~0.006. Assuming that the IEPOX-OA factor would have an $f_{82}$ value of 0.0132 based on the work of Budisulistiorini et al. (2013), the contribution of the IEPOX-OA to total OA in the present study would be at most 0.5%. It is considered infeasible to extract factors with such low contributions to total OA mass; for example, Ulbrich et al. (2009) suggest that only factors which contribute at least 5% to total OA mass can be extracted reliably using PMF."

By this line of logic, the maximum IEPOX-OA contribution would be 50%, which is unrealistic. We note that the observed $f_{82}$ value (5.5 – 6.4 ‰) is lower than the average value observed for ambient OA that has been influenced by isoprene emissions (6.5 ± 2.2 ‰) (Hu et al., 2015), which would suggest that IEPOX-OA did not contribute significantly to SOA formation (Wong et al., 2015). In other chlorine-initiated SOA formation from biogenic volatile organic compounds experiments, the observed $f_{82}$ values was in some cases as high as 5 ‰, which is close to observe $f_{82}$ values for isoprene-chlorine SOAs. As shown in Figure 6 and 7, $C_5H_8O_2$, which could produce $C_5H_6O^+$ fragment within the ACSM, was observed as a gas-phase product. It is plausible that other biogenic SOA species would produce ion fragments at $m/z$ 82 ($C_5H_6O^+$ or other ions with the same integer $m/z$). A more detailed discussion is included in the revised manuscript. We have included the UMR ACSM spectra comparison here, which can also be found in the updated S.I (section S6).

Section 3.4, Pg.11 Line 32 – Pg.14 Line 13: "Another way to test the presence of IEPOX is to reduce aerosol pH, which should lead to increased uptake of IEPOX (Budisulistiorini et al., 2013; Gaston et al., 2014; Hu et al., 2015; Lin et al., 2012; Riedel et al., 2016, 2015). Comparison of ACSM mass spectra (see Fig. S13) suggests that the presence of acidic aerosol increases the contribution of ion mass fragments at $m/z$ 82 ($C_5H_6O^+$, "$f_{82}$") to the overall SOA mass, which is associated with IEPOX-derived OA (Budisulistiorini et al., 2013; Hu et al., 2015). However, the magnitude of change is low (1 ‰) and within uncertainty of the instrument. Interference by non-IEPOX-derived OA fragments and non-$C_5H_6O^+$ ions at $m/z$ 82 is also possible. Separate monoterpene-chlorine experiments observed $f_{82}$ values as high as 5 ‰. The observed $f_{82}$ values for isoprene-chlorine SOA are below the average value observed for ambient OA influenced by isoprene emission (6.5 ± 2.2 ‰) and much lower than IEPOX-derived SOA (12-40 ‰) observed in laboratory studies (Hu et al., 2015). We also attempted to but were unable to extract an IEPOX factor using positive matrix factorization (Ulbrich et al., 2009), as some studies have done (Budisulistiorini et al., 2013; Lin et al., 2012). Reduction in gas-phase products including those resembling IEPOX was also observed in the CIMS when the aerosol was acidic (see Fig. S14). These observations are consistent with increased partitioning of gas-phase products to the aerosol when the seed aerosol is acidic, resulting in the higher SOA concentrations shown in Fig. 1 and Table 1, but do not prove the presence of IEPOX-derived SOA."

[Figure]

**Figure S13.** Comparison of ACSM unit mass spectra. Red bars indicate *m/z* fragments enhanced in the presence of acidic aerosols. Green bars indicate *m/z* fragments enhanced in the presence of neutral aerosols.

3.) Reviewer: *Please go through carefully and make sure certain references are not missing throughout the text. I mention a few of these in my minor comments below.*

Response: Additional references have been included.

4.) Reviewer: *I know it isn't a focus in this manuscript, but it would be very powerful if molecular tracers could be identified for Cl-initiated radicals yielding SOA. The authors mention using the ACSM to try to constrain the organochlorine budget, but seemed to have trouble with this due to interference issues. This is why I suggested conducting PMF/ME-2 analyses above in # 2. However, does the CIMS data (especially the iodide reagent ion chemistry) suggest the presence of low-volatility hydroperoxides that contain chlorine in them? From OH radical studies by Krechmer et al. (2015, ES&T), Riva et al. (2016, ES&T), and Liu et al. (2016, ES&T), they all measured low-volatility multifunctional hydroperoxides that made sufficient amounts of SOA (that don't require aerosol acidity like IEPOX).*

Response: Based on our CIMS measurements, multigenerational reaction pathways that could lead to hydroperoxide formation from continued oxidation of early $C_5$ oxidation products are plausible. We show time series for some of these multifunctional chloroalkyl hydroperoxides in the updated Figure 6 and added a summary of proposed products in Figure 7. More detailed formation pathways are added as well to the S.I. (see Fig S10-12). References to previous work on hydroperoxide formation/oxidation under low $NO_x$ are added as well.

Figure 6a and 6b: Updated with hydroperoxides, chloroalkyl hydroperoxides and additional multi-generational products observed by CIMS.

Figure 7: Observations by $(H_2O)_nI^-$ are now shown in Fig. 6b. Figure 7 now summarizes proposed reaction products, some of which are (chlorinated) hydroperoxides.

Section 3.4, Pg.11, Line 3-8: "For instance, the $C_5H_8ClO_2^{\bullet}$ radical produced via Cl-addition to isoprene could either undergo $RO_2 + RO_2$ chemistry to produce $C_5H_7ClO$ (e.g. CMBO) or undergo $RO_2 + HO_2$ chemistry to produce $C_5H_9ClO_2$, a chloroalkyl hydroperoxide. Similarly, the $C_5H_7O_2^{\bullet}$ radical produced via H-abstraction from isoprene could undergo $RO_2 + RO_2$ chemistry to produce $C_5H_6O$ (e.g. MBO) or undergo $RO_2 + HO_2$ chemistry to produce $C_5H_8O_2$, a hydroperoxide. Ions consistent with $C_5H_9ClO_2$ and $C_5H_8O_2$ are observed in the gas-phase, as shown in Fig 6a and 6b, where the formation of $RO_2 + HO_2$ reaction products appears delayed compared to their $RO_2 + RO_2$ pathway counterparts."

Section 3.4, Pg.11, Line 11-14: "In the OH-isoprene system, multifunctional, low volatility hydroperoxides produced from non-IEPOX ("isoprene-derived epoxydiol") reaction pathways contribute to SOA formation under low $NO_x$ conditions (Krechmer et al., 2015; Liu et al., 2016; Riva et al., 2016). Analogously, in the Cl-isoprene system, the (chloroalkyl) hydroperoxide species identified in Fig. 6 and Fig. 7 are expected to contribute to SOA formation."

(5). Reviewer: *When reviewing Table 1, I realized it wasn't well explained in the text why the different injection methods were used. What did these methods explicitly tell you?*

Response: The different injection methods were used to separate SOA formation from the effects of vapor wall loss. We have added some clarifying text.

Section 2.1, Pg.4, Line 10-13: "Initial $Cl_2$ experiments were performed to achieve rapid oxidation of isoprene and to separate initial SOA formation from effects of vapor wall loss. Because chlorine radicals are not expected to regenerate, continuous $Cl_2$ injection experiments were performed to provide more steady but lower Cl radical concentrations."

Section 3.1, Pg.7, Line 3-6: "The initial chlorine experiment (A4) exhibited similar trends, but the SOA decay was faster, where 30 µg m$^{-3}$ SOA decay (40% of maximum SOA mass) occurred within about 30 minutes of photo-oxidation ($9 < T < 40$ mins), likely due to more rapid oxidation and fragmentation of reaction products."

Section 3.4, Pg.9, Line 20-22: "In initial chlorine injection experiments, maximum SOA concentrations were reached within 15 minutes, and the effects of vapor wall loss, oxidative fragmentation, and photolysis on reported maximum SOA yield were lower than during continuous chlorine injection experiments."

5   *Reviewer: For modelers, this Table might be very difficult for them to judge which yields should be used. Also, related to my point # 1 above, modelers seeing these yields may question if these yields are accurate to remote low-NOx regions where Cl radical chemistry might matter. Can the authors offer which yield may be the most appropriate to use?*

Response: We have added the following qualifications and recommendations:

Section 3.3, Pg.9, Line 23-32: "Under atmospheric conditions, the isoprene-chlorine ratio will usually be higher than ratios used in these experiments. Previous studies on chlorine-initiated SOA formation from toluene (Cai et al., 2008) and limonene (Cai and Griffin, 2006) suggest that SOA yields decrease with higher VOC-to-Cl ratio. While we do not observe a clear correlation between SOA yield and isoprene-to-chlorine ratios used here (0.5-1.2), such dependence could be present over a

15   wider ratio range.  For air quality models which do not explicitly account for fragmentation reactions, the use of the average continuous case yield, which is similar to recently reported OH-oxidation yields  (Liu et al., 2016; Xu et al., 2014), is more appropriate because the isoprene-to-chlorine ratio is closer to atmospheric conditions and because the SOA yields from continuous injection experiments account for effects of OA aging in the atmosphere (which occur throughout the experiments). The presence of acidic aerosols and inclusion of particulate chlorine content would increase expected yields."

*Reviewer: Finally, I'm assuming these various injection methods were used to gain some insights into vapor wall losses? It remains unclear to me how exactly vapor wall losses were dealt with (if at all) in reporting the SOA yields shown in Table 1.*

Response: The different injection methods were indeed used to separate the effects of vapor wall loss from SOA formation, as

25   clarified above. The wall loss correction method used accounts for depositional particle loss, as well as organic vapor loss to the deposited particles. Essentially, the correction method assumes that organic aerosols lost to the chamber wall would still participate in equilibrium partitioning as if they were suspended. Loss of organic vapor to the clean Teflon® surface is not accounted for here. This was described in more concise terms in the manuscript,

30   Section 2.3, Pg.5, Line 11-14: "Assuming internal mixing of particles and that organic vapor can condense onto suspended and wall-deposited particles alike, we corrected for particle wall loss and the loss of organic vapors onto wall-deposited particles using the organic-to-sulfate ratio (Hildebrandt et al., 2009)"

Section 2.3, Pg.5, Line 17-18: "This correction does not account for loss of organic vapors to clean Teflon® surfaces."

Minor comments and responses.

*1.) Reviewer: Abstract, Page 1, Line 12: Remove "%" after "8." You don't need this.*

Response: We have removed "%"

*2.) Reviewer: Methods, Page 2, Line 32: Insert a space between "exceeding100"*

Response: We have fixed this typographical error.

*3.) Reviewer: Section 3.4, Page 7, Line 27: You write "3-methyl-3-butene-2-one (CMBO) [(C5H7OCl)H+]." This appears to*

10 *be named incorrectly. Please name according to IUPAC.*

Response: We have corrected the naming to 1-chloro-3-methyl-3-butene-2-one (CMBO) in Section 1, Pg.3, Line 4. The abbreviation "CMBO" is now used throughout section 3.4

*4.) Reviewer: Section 3.4, Page 7, Line 31: I would reference Kroll et al. (2006, ES&T) and Surratt et al. (2006, JPCA) as one*

15 *of the initial references to demonstrates MACR oxidation is a source of SOA.*

Response: We have modified the references as suggested. See section 3.4, Pg.10, Line 6-7.

*5.) Reviewer: Section 3.4, Page 8, Line 26: The authors should reference Lin et al. (2012, ES&T)*

Response: We have added the suggested reference. See section 3.4, Pg.11, Line 17-18.

*6.) Reviewer: Section 3.4, Page 8, Line 27: The authors should also reference Gaston et al. (2014, ES&T) and Riedel et al.*
*(2015, ES&T Letters)*

Response: We have added the suggested reference. See also section 3.4, Pg.11, Line 17-18.

[revised manuscript text omitted]